# Wasserstein Flow Meets Replicator Dynamics: A Mean-Field Analysis of Representation Learning in Actor-Critic

**Yufeng Zhang**†
Northwestern University
yufengzhang2023@u.northwestern.edu

**Siyu Chen**†
Tsinghua University
chensy18@mails.tsinghua.edu.cn

**Zhuoran Yang**
Princeton University
zy6@princeton.edu

**Michael I. Jordan**
UC Berkeley
jordan@cs.berkeley.edu

**Zhaoran Wang**
Northwestern University
zhaoranwang@gmail.com

## Abstract

Actor-critic (AC) algorithms, empowered by neural networks, have had significant empirical success in recent years. However, most of the existing theoretical support for AC algorithms focuses on the case of linear function approximations, or linearized neural networks, where the feature representation is fixed throughout training. Such a limitation fails to capture the key aspect of representation learning in neural AC, which is pivotal in practical problems. In this work, we take a mean-field perspective on the evolution and convergence of feature-based neural AC. Specifically, we consider a version of AC where the actor and critic are represented by overparameterized two-layer neural networks and are updated with two-timescale learning rates. The critic is updated by temporal-difference (TD) learning with a larger stepsize while the actor is updated via proximal policy optimization (PPO) with a smaller stepsize. In the continuous-time and infinite-width limiting regime, when the timescales are properly separated, we prove that neural AC finds the globally optimal policy at a sublinear rate. Additionally, we prove that the feature representation induced by the critic network is allowed to evolve within a neighborhood of the initial one.

## 1 Introduction

In reinforcement learning (RL) [56], an agent aims to learn the optimal policy that maximizes the expected total reward by interacting with the environment. Policy-based RL algorithms achieve such a goal by directly optimizing the expected total reward as a function of the policy, which often involves two components: policy evaluation and policy improvement. Specifically, policy evaluation refers to estimating the value function of the current policy, which characterizes the performance of the current policy and reveals the updating direction for finding a better policy, which is known as policy improvement. Algorithms with these two ingredients are also called actor-critic (AC) methods [36], where the actor and the critic refer to the policy and its corresponding value function, respectively.

---

† Equal contribution.

35th Conference on Neural Information Processing Systems (NeurIPS 2021).

Recently, in RL applications with large state spaces, actor-critic empowered by expressive function approximators such as neural networks have achieved striking empirical successes [3, 4, 9, 20, 51, 52, 60]. These successes benefit from the data-dependent representations learned by neural networks. Unfortunately, however, the theoretical understanding of this data-dependent benefit is very limited. The classical theory of AC focuses on the case of linear function approximation, where the actor and critic are represented using linear functions with the feature mapping fixed throughout learning [10, 11, 36]. Meanwhile, a few recent works establish convergence and optimality of AC with overparameterized neural networks [26, 39, 61], where the neural network training is captured by the Neural Tangent Kernel (NTK) [30]. Specifically, with properly designed parameter initialization and stepsizes, and sufficiently large network widths, the neural networks employed by both actor and critic can be assumed to be well approximated by linear functions of a random feature determined by initial parameters. In other words, concerning representation learning, the features induced by these algorithms are by assumption infinitesimally close to the initial featural representation, which is data-independent.

In this work, we make initial steps towards understanding how representation learning comes into play in neural AC. Specifically, we address the following questions:

*Going beyond the NTK regime, does neural AC provably find the globally optimal policy? How does the feature representation associated with the neural network evolve along with neural AC?*

We focus on a version of AC where the critic performs temporal-difference (TD) learning [55] for policy evaluation and the actor improves its policy via proximal policy optimization (PPO) [49], which corresponds to a Kullback-Leibler (KL) divergence regularized optimization problem, with the critic providing the update direction. Moreover, we utilize two-timescale updates where both the actor and critic are updated at each iteration but with the critic having a much larger stepsize. In other words, the critic is updated at a faster timescale. Meanwhile, we represent the critic explicitly as a two-layer overparameterized neural network and parameterize the actor implicitly via the critic and PPO updates. To examine convergence, we study the evolution of the actor and critic in the continuous-time limiting regime with the network width going to infinity. In such a regime, the actor update is closely connected to replicator dynamics [12, 28, 50] and the critic update is captured by a semigradient flow in the Wasserstein space [59]. Moreover, the semigradient flow runs at a faster timescale according to the two-timescale mechanism.

It turns out that the separation of timescales plays an important role in the convergence analysis. In particular, in the continuous-time limit, it enables us to first separately analyze the evolution of actor and critic and then combine these results to get final theoretical guarantees. Specifically, focusing solely on the actor, we prove that the time-averaged suboptimality of the actor converges sublinearly to zero up to the time-averaged policy evaluation error associated with critic updates. Moreover, for the critic, under proper regularity conditions, we connect the Bellman error to the Wasserstein distance and show that the time-averaged policy evaluation error also converges sublinearly to zero. Therefore, we show that neural AC provably achieves global optimality at a sublinear rate. Furthermore, regarding representation learning, we show that the critic induces a data-dependent feature representation within an $O(1/\alpha)$ neighborhood of the initial representation in terms of the Wasserstein distance, where $\alpha$ is a sufficiently large scaling parameter.

The key to our technical analysis reposes on three ingredients: (i) infinite-dimensional variational inequalities with a one-point monotonicity [27], (ii) a mean-field perspective on neural networks [19, 41, 42, 53, 54], and (iii) the two-timescale stochastic approximation [13, 37]. In particular, in the infinite-width limit, the neural network and its induced feature representation are identified with a distribution over the parameter space. The mean-field perspective enables us to characterize the evolution of such a distribution within the Wasserstein space via a partial differential equation (PDE) [5, 6, 58, 59]. For policy evaluation, such a PDE is given by the semigradient flow induced by TD learning. We characterize the error of policy evaluation by showing that mean-field Bellman error satisfies a version of one-point monotonicity tailored to the Wasserstein space. Moreover, our actor analysis utilizes the geometry of policy optimization, which shows that the expected total reward,

as a function of the policy, also enjoys the property of one-point monotonicity in the policy space. Finally, the actor and critic errors are connected via two-timescale stochastic approximation. To the best of our knowledge, this is the first time that convergence and global optimality guarantees have been obtained for neural AC.

**Related Work.** AC with linear function approximation has been studied extensively in the literature. In particular, using a two-timescale stochastic approximation via ordinary differential equations, [10, 11, 36] establish asymptotic convergence guarantees in the continuous-time limiting regime. More recently, using more sophisticated optimization techniques, various works [29, 35, 64–66] have established discrete-time convergence guarantees that show that linear AC converges sublinearly to either a stationary point or the globally optimal policy. Furthermore, when overparameterized neural networks are employed, [26, 39, 61] prove that neural AC converges to the global optimum at a sublinear rate. In these works, the initial value of the network parameters and the learning rates are chosen such that both actor and critic updates are captured by the NTK. In other words, when the network width is sufficiently large, such a version of neural AC is well approximated by its linear counterpart via the neural tangent feature. In comparison, we establish a mean-field analysis that has a different scaling than the NTK regime. We also establish finite-time convergence to global optimality, and more importantly, the feature representation induced by the critic is data-dependent and allowed to evolve within a much larger neighborhood around the initial one.

Furthermore, our work is also related to the recent line of research on understanding stochastic gradient descent (SGD) for supervised learning problems involving an overparameterized two-layer neural network under the mean-field regime. See, e.g., [16, 19, 21, 22, 31, 40–42, 53, 54, 63] and the references therein. In the continuous-time and infinite-width limit, these works show that SGD for neural network training is captured by a Wasserstein gradient flow [5, 6, 59] of an energy function that corresponds to the objective function in supervised learning. In contrast, our analysis combines such a mean-field analysis with TD learning and two-timescale stochastic approximation, which are tailored specifically to AC. Moreover, our critic is updated via TD learning, which is a semigradient algorithm and there is no objective functional making TD learning a gradient-based algorithm. Thus, in the mean-field regime, our critic is given by a Wasserstein semigradient flow, which also differs from these existing works.

Additionally, our work is closely related to [1, 69], who provide mean-field analyses for neural TD-learning and Q-learning [62]. In comparison, we focus on AC, which is a two-timescale policy optimization algorithm. Finally, [2] studies softmax policy gradient with neural network policies in the mean-field regime, where policy gradient is cast as a Wasserstein gradient flow with respect to the expected total reward. The algorithm assumes that the critic directly gets the desired value function and thus the algorithm is single-timescale. Moreover, the convergence guarantee in [2] is asymptotic. In comparison, our AC is two-timescale and we establish non-asymptotic sublinear convergence guarantees to global optimality.

**Notation.** We denote by $\mathscr{P}(\mathcal{X})$ the set of probability measures over the measurable space $\mathcal{X}$. Given a curve $\rho : \mathbb{R} \to \mathcal{X}$, we denote by $\dot{\rho}_s = \partial_t \rho_t \mid_{t=s}$ its derivative with respect to the time. For an operator $F : \mathcal{X} \to \mathcal{X}$ and a measure $\mu \in \mathscr{P}(\mathcal{X})$, we denote by $F_\sharp \mu = \mu \circ F^{-1}$ the push forward of $\mu$ through $F$. We denote by $\chi^2(\rho \| \mu)$ the chi-squared divergence between probability measures $\rho$ and $\mu$, which is defined as $\chi^2(\rho \| \mu) = \int (\rho/\mu - 1)^2 \mathrm{d}\mu$. Given two probability measures $\rho$ and $\mu$, we denote the Kullback-Leibler divergence or the relative entropy from $\mu$ to $\rho$ by $\mathrm{KL}(\rho \| \mu) = \int \log(\rho/\mu) \mathrm{d}\rho$. For $\nu_1, \nu_2, \mu \in \mathscr{P}(\mathcal{X})$, we define the $\dot{H}^{-1}(\mu)$ weighted homogeneous Sobolev norm as $\|\nu_1 - \nu_2\|_{\dot{H}^{-1}(\mu)} = \sup \left\{ |\langle f, \nu_1 - \nu_2 \rangle| \big| \|f\|_{\dot{H}^1(\mu)} \leq 1 \right\}$. We denote by $\|f(x)\|_{p,\mu} = (\int |f(x)|^p \mu(\mathrm{d}x))^{1/p}$ the $\ell_p$-norm with respect to probability measure $\mu$. We denote by $\otimes$ the semidirect product, i.e., $\mu \otimes K = K(y \mid x)\mu(x)$ for $\mu \in \mathscr{P}(\mathcal{X})$ and transition kernel $K : \mathcal{X} \to \mathscr{P}(\mathcal{Y})$. For a function $f : \mathcal{X} \to \mathbb{R}$, we denote by $\mathrm{Lip}(f) = \sup_{x,y \in \mathcal{X}, x \neq y} |f(x) - f(y)|/\|x - y\|$ its Lipschitz constant. We denote a normal distribution on $\mathbb{R}^D$ by $\mathcal{N}(\mu, \Sigma)$, where $\mu$ is the mean value and $\Sigma$ is the covariance matrix.

## 2 Background

In this section, we first introduce the policy optimization problem and the actor-critic method. We then present the definition of the Wasserstein space.

### 2.1 Policy Optimization and Actor-Critic

We consider a Markov decision process (MDP) given by $(\mathcal{S}, \mathcal{A}, \gamma, P, r, \mathcal{D}_0)$, where $\mathcal{S} \subseteq \mathbb{R}^{d_1}$ is the state, $\mathcal{A} \subseteq \mathbb{R}^{d_2}$ is the action space, $\gamma \in (0, 1)$ is the discount factor, $P : \mathcal{S} \times \mathcal{A} \to \mathscr{P}(\mathcal{S})$ is the transition kernel, $r : \mathcal{S} \times \mathcal{A} \to \mathbb{R}_+$ is the reward function, and $\mathcal{D}_0 \in \mathscr{P}(\mathcal{S})$ is the initial state distribution. Without loss of generality, we assume that $\mathcal{S} \times \mathcal{A} \subseteq \mathbb{R}^d$ and $\|(s, a)\|_2 \leq 1$, where $d = d_1 + d_2$. We remark that as long as the state-action space is bounded, we can normalize the space to be within the unit sphere. Given a policy $\pi : \mathcal{S} \times \mathcal{A} \to \mathscr{P}(\mathcal{S})$, at the $m$th step, the agent takes an action $a_m$ at state $s_m$ according to $\pi(\cdot \,|\, s_m)$ and observes a reward $r_m = r(s_m, a_m)$. The environment then transits to the next state $s_{m+1}$ according to the transition kernel $P(\cdot \,|\, s_m, a_m)$. Note that the policy $\pi$ induces Markov chains on both $\mathcal{S}$ and $\mathcal{S} \times \mathcal{A}$. Considering the Markov chain on $\mathcal{S}$, we denote the induced Markov transition kernel by $P^\pi : \mathcal{S} \to \mathscr{P}(\mathcal{S})$, which is defined as $P^\pi(s' \,|\, s) = \int_{\mathcal{A}} P(s' \,|\, s, a)\pi(\mathrm{d}a \,|\, s)$. Likewise, we denote the Markov transition kernel on $\mathcal{S} \times \mathcal{A}$ by $\widetilde{P}^\pi : \mathcal{S} \times \mathcal{A} \to \mathscr{P}(\mathcal{S} \times \mathcal{A})$, which is defined as $\widetilde{P}^\pi(s', a' \,|\, s, a) = \pi(a' \,|\, s')P(s' \,|\, s, a)$. Let $\widetilde{\mathcal{D}}$ be a probability measure on $\mathcal{S} \times \mathcal{A}$. We then define the visitation measure induced by policy $\pi$ and starting from $\widetilde{\mathcal{D}}$ as

$$\widetilde{\mathcal{E}}^\pi_{\widetilde{\mathcal{D}}}\big(\mathrm{d}(s, a)\big) = (1 - \gamma) \cdot \sum_{m \geq 0} \gamma^m \cdot \mathbb{P}\big((s_m, a_m) \in \mathrm{d}(s, a) \,|\, (s_0, a_0) \sim \widetilde{\mathcal{D}}\big), \qquad (2.1)$$

where $(s_m, a_m)$ is the trajectory generated by starting from $(s_0, a_0) \sim \widetilde{\mathcal{D}}$ and following policy $\pi$ thereafter. If $\widetilde{\mathcal{D}} = \mathcal{D} \otimes \pi$ holds, we then denote such a visitation measure by $\widetilde{\mathcal{E}}^\pi_{\mathcal{D}}$. Furthermore, we denote by $\mathcal{E}(\mathrm{d}s) = \int_{\mathcal{A}} \widetilde{\mathcal{E}}(\mathrm{d}s, \mathrm{d}a)$ the marginal distribution of visitation measure $\widetilde{\mathcal{E}}$ with respect to $\mathcal{S}$. In particular, when $(s_0, a_0) \sim \mathcal{D} \otimes \pi$ holds in (2.1), it follows that $\widetilde{\mathcal{E}}^\pi_{\mathcal{D}} = \mathcal{E}^\pi_{\mathcal{D}} \otimes \pi$. In policy optimization, we aim to maximize the expected total rewards $J(\pi)$ defined as follows,

$$J(\pi) = \mathbb{E}^\pi \left[ \sum_{m \geq 0} \gamma^m \cdot r(s_m, a_m) \,\Big|\, s_0 \sim \mathcal{D}_0 \right],$$

where we denote by $\mathbb{E}^\pi$ the expectation with respect to $a_m \sim \pi(\cdot \,|\, s_m)$ and $s_{m+1} \sim P(\cdot \,|\, s_m, a_m)$ for $m \geq 0$. We define the action value function $Q^\pi : \mathcal{S} \times \mathcal{A} \to \mathbb{R}$ and the state value function $V^\pi : \mathcal{S} \to \mathbb{R}$ as follows,

$$Q^\pi(s, a) = \mathbb{E}^\pi \left[ \sum_{m \geq 0} \gamma^m \cdot r(s_m, a_m) \,\Big|\, s_0 = s, a_0 = a \right], \quad V^\pi(s) = \big\langle Q^\pi(s, \cdot), \pi(\cdot \,|\, s) \big\rangle_{\mathcal{A}}, \quad (2.2)$$

where we denote by $\langle \cdot, \cdot \rangle_{\mathcal{A}}$ the inner product on the action space $\mathcal{A}$. Correspondingly, the advantage function $A^\pi : \mathcal{S} \times \mathcal{A} \to \mathbb{R}$ is defined as

$$A^\pi(s, a) = Q^\pi(s, a) - V^\pi(s).$$

It is known that the action value function $Q^\pi$ is the unique global minimizer to the following mean-squared Bellman error (MSBE),

$$\mathrm{MSBE}(Q; \pi) = \frac{1}{2}\mathbb{E}_{(s,a) \sim \widetilde{\Phi}^\pi} \left[ \big(Q(s, a) - r(s, a) - \gamma\mathbb{E}_{(s',a') \sim \widetilde{P}^\pi(\cdot \,|\, s, a)}[Q(s', a')]\big)^2 \right], \qquad (2.3)$$

where $\widetilde{\Phi}^\pi$ is a weighting distribution depending on policy $\pi$ and is with full support, i.e., $\mathrm{supp}(\widetilde{\Phi}^\pi) = \mathcal{S} \times \mathcal{A}$. Therefore, the policy optimization problem can be written as the following bilevel optimization problem,

$$\max_\pi J(\pi) = \mathbb{E}_{s \sim \mathcal{D}_0} \Big[ \big\langle Q^\pi(s, \cdot), \pi(\cdot \,|\, s) \big\rangle_{\mathcal{A}} \Big], \quad \text{subject to } Q^\pi = \operatorname*{argmin}_Q \mathrm{MSBE}(Q; \pi). \qquad (2.4)$$

The inner problem in (2.4) is known as a policy evaluation subproblem, while the outer problem is the policy improvement subproblem. One of the most popular way to solve the policy optimization problem is actor-critic (AC) methods [56], where the job of the critic is to evaluate current policy and then the actor updates its policy according to the critic's evaluation.

## 2.2 Wasserstein Space

Let $\Theta \subseteq \mathbb{R}^D$ be a Polish space. We denote by $\mathscr{P}_2(\Theta) \subseteq \mathscr{P}(\Theta)$ the set of probability measures with finite second moments. Then, the Wasserstein-2 ($W_2$) distance between $\mu, \nu \in \mathscr{P}_2(\Theta)$ is defined as follows,

$$W_2(\mu, \nu) = \inf\left\{\mathbb{E}\big[\|X - Y\|^2\big]^{1/2} \,\Big|\, \text{law}(X) = \mu, \text{law}(Y) = \nu\right\},$$

where the infimum is taken over the random variables $X$ and $Y$ on $\Theta$ and we denote by $\text{law}(X)$ the distribution of a random variable $X$. We call $\mathcal{M} = (\mathscr{P}_2(\Theta), W_2)$ the Wasserstein ($W_2$) space, which is an infinite-dimensional manifold [59]. See §A.1 for more details.

## 3 Algorithm

**Two-timescale Actor-critic.** We consider a two-timescale Actor-critic (AC) algorithm [34, 45] for the policy optimization problem in (2.4). For policy evaluation, we parameter the critic $Q$ with a neural network and update the parameter via temporal-difference (TD) learning [55]. For policy improvement, we update the actor policy $\pi$ via proximal policy optimization (PPO) [49]. Our algorithm is two-timescale since both the actor and critic are updated at each iteration with different stepsizes. Specifically, we parameterize the critic $Q$ by the following neural network with width $M$ and parameter $\widehat{\boldsymbol{\theta}} = (\widehat{\theta}^{(1)}, \cdots, \widehat{\theta}^{(M)}) \in \mathbb{R}^{D \times M}$,

$$Q_{\widehat{\boldsymbol{\theta}}}(s, a) = \frac{\alpha}{M} \sum_{i=1}^{M} \sigma(s, a; \widehat{\theta}^{(i)}). \tag{3.1}$$

Here $\sigma(s, a; \theta) : \mathcal{S} \times \mathcal{A} \times \mathbb{R}^D \to \mathbb{R}$ is the activation function and $\alpha > 0$ is the scaling parameter. Such a structure also appears in [17, 18, 41]. In a discrete-time finite-width (DF) scenario, at the $k$th iteration, the critic and actor are updated as follows,

DF-TD: $\quad \widehat{\theta}_{k+1}^{(i)} = \widehat{\theta}_k^{(i)} - \frac{\varepsilon'}{\alpha}\big(Q_{\widehat{\boldsymbol{\theta}}_k}(s_k, a_k) - r(s_k, a_k) - \gamma Q_{\widehat{\boldsymbol{\theta}}_k}(s_k', a_k')\big)\nabla_\theta \sigma(s, a; \widehat{\theta}_k^{(i)}),$ (3.2)

DF-PPO: $\quad \widehat{\pi}_{k+1}(\cdot\,|\,s) = \underset{\pi}{\arg\max}\Big\{\big\langle Q_{\widehat{\boldsymbol{\theta}}_k}(s, \cdot), \pi(\cdot\,|\,s)\big\rangle_{\mathcal{A}} - \varepsilon^{-1} \cdot \text{KL}\big(\pi(\cdot\,|\,s)\,\|\,\widehat{\pi}_k(\cdot\,|\,s)\big)\Big\},$ (3.3)

where $(s_k, a_k) \sim \widetilde{\Phi}^{\widehat{\pi}_k}$ and $(s_k', a_k') \sim \widetilde{P}^{\widehat{\pi}_k}(\cdot\,|\,s_k, a_k)$. Here $\widehat{\pi}_k$ is the policy for the actor at the $k$th iteration, $\widetilde{\Phi}^{\widehat{\pi}_k}$ is the corresponding weighting distribution, $\varepsilon$ and $\varepsilon'$ are the stepsizes for the DF-PPO update and the DF-TD update, respectively. In (3.2), the scaling of $\alpha^{-1}$ arises since our update falls into the lazy-training regime [18]. In the sequel, we denote by $\eta = \varepsilon'/\varepsilon$ the relative TD timescale. Note that in a double-loop AC algorithm, the critic can usually be solved with high precision. In the two-timescale AC however, even with the KL-divergence term in (3.3) which regularizes the policy update and helps to improve the local estimation quality of the TD update, the critic $Q_{\widehat{\boldsymbol{\theta}}_k}$ for updating the actor's policy $\widehat{\pi}_k$ can still be far from the true action value function $Q^{\widehat{\pi}_k}$. Since the policy evaluation problem is not fully solved at each iteration, the two-timescale AC can be more efficient in computation while more challenging to establish a theoretical guarantee.

**Mean-field (MF) Limit.** To analyze the convergence of the two-timescale AC with neural networks, we employ the analysis that studies the mean-field limit regime [41, 42]. Here, by saying the mean-field limit, we refer to the infinite-width limit, i.e., $M \to \infty$ for the neural network width $M$ in (3.1), and the continuous-time limit, i.e., $t = k\varepsilon$ where $\varepsilon \to 0$ for the stepsize in (3.2) and (3.3). For $\widehat{\boldsymbol{\theta}} = \{\widehat{\theta}^{(i)}\}_{i=1}^{M}$ independently sampled from a distribution $\rho$, we can write the infinite-width limit of (3.1) as

$$Q(s, a; \rho) = \alpha \int \sigma(s, a; \theta)\rho(\mathrm{d}\theta). \tag{3.4}$$

In the sequel, we denote by $\widehat{\rho}_k$ the distribution of $\widehat{\theta}_k^{(i)}$ for the infinite-width limit of the neural network at the $k$th iteration. We further let $\rho_t$ and $\pi_t$ be the continuous-time limits of $\widehat{\rho}_k$ and $\widehat{\pi}_k$, respectively.

As studied in [69], the mean-field limit of the DF-TD update in (3.2) is

$$\text{MF-TD:} \quad \partial_t \rho_t = -\eta \operatorname{div}\big(\rho_t \cdot g(\cdot; \rho_t, \pi_t)\big), \tag{3.5}$$

where $\eta$ is the relative TD timescale and

$$g(\theta; \rho, \pi) = -\mathbb{E}^{\pi}_{\widetilde{\Phi}^{\pi}}\Big\{\big[Q(s,a;\rho) - r(s,a) - \gamma \cdot Q(s',a';\rho)\big] \cdot \alpha^{-1}\nabla_\theta \sigma(s,a;\theta)\Big\} \tag{3.6}$$

is a vector field. Here $\mathbb{E}^{\pi}_{\widetilde{\Phi}^{\pi}}$ is taken with respect to $(s,a) \sim \widetilde{\Phi}^{\pi}$ and $(s',a') \sim \widetilde{P}^{\pi}(\cdot \mid s,a)$. It remains to characterize the mean-field limit of the DF-PPO update in (3.3). By solving the maximization problem in (3.3), the infinite-width limit of DF-PPO update can be written in closed form as

$$\varepsilon^{-1} \cdot \Big\{\log\big[\widehat{\pi}_{k+1}(a \mid s)\big] - \log\big[\widehat{\pi}_k(a \mid s)\big]\Big\} = Q(s,a;\widehat{\rho}_k) - \widehat{Z}_k(s),$$

where $\widehat{Z}_k(s)$ is the normalizing factor such that $\int \widehat{\pi}_k(\mathrm{d}a \mid s) = 1$ for any $s \in \mathcal{S}$. By letting $t = k\varepsilon$ and $\varepsilon \to 0$, we have $\partial_t \log \pi_t = Q_t - Z_t$, which can be further written as $\partial_t \pi_t = \pi_t \cdot (Q_t - Z_t)$. Here we have $Q_t(a,s) = Q(a,s;\rho_t)$ and $Z_t$ is the continuous-time limit of $\widehat{Z}_k$. Furthermore, noting that $\partial_t \int \pi_t(\mathrm{d}a \mid s) = 0$, the mean-field limit of the DF-PPO update in (3.3) is

$$\text{MF-PPO:} \quad \frac{\mathrm{d}}{\mathrm{d}t}\pi_t = \pi_t \cdot A_t, \qquad \text{where } A_t(s,a) = Q_t(s,a) - \int Q_t(s,a)\pi_t(\mathrm{d}a \mid s). \tag{3.7}$$

The two updates (3.5) and (3.7) correspond to the mean-field limits of (3.2) and (3.3), respectively, and together serve as the mean-field limit of the two-timescale AC. In particular, we remark that the MF-TD update in (3.5) for the critic is captured by a semigradient flow in the Wasserstein space [59] while the MF-PPO update in (3.7) for the actor resembles the replicator dynamics [12, 28, 50]. Note that such a framework is applicable to continuous state and action space. In this paper, we aim to provide a theoretical analysis of the mean-field limit of the two-timescale AC.

## 4 Main Result

In this section, we first establish the convergence of the MF-PPO update in §4.1. Then, with additional assumptions, we establish the optimality and convergence of the mean-field two-timescale AC in §4.2.

### 4.1 Convergence of Mean-field PPO

For the MF-PPO update in (3.7), we establish the following theorem on its global optimality and convergence rate.

**Theorem 4.1** (Convergence of MF-PPO). Let $\pi^* = \operatorname{argmax}_\pi J(\pi)$ be the optimal policy and $\pi_0$ be the initial policy. Then, it holds that

$$\frac{1}{T}\int_0^T \big(J(\pi^*) - J(\pi_t)\big)\mathrm{d}t \le \frac{\zeta}{T} + 4\kappa \cdot \underbrace{\frac{1}{T}\int_0^T \|Q_t - Q^{\pi_t}\|_{2,\widetilde{\phi}^{\pi_t}}\mathrm{d}t}_{\text{policy evaluation error}}, \tag{4.1}$$

where $\widetilde{\phi}^{\pi_t} \in \mathscr{P}(\mathcal{S} \times \mathcal{A})$ is an evaluation distribution for the policy evaluation error and $\zeta = \mathbb{E}_{s \sim \mathcal{E}^{\pi^*}_{\mathcal{D}_0}}\big[\operatorname{KL}\big(\pi^*(\cdot \mid s) \,\|\, \pi_0(\cdot \mid s)\big)\big]$ is the expected KL-divergence between $\pi^*$ and $\pi_0$. Furthermore, letting $\widetilde{\phi}^{\pi_t} = \frac{1}{2}\widetilde{\phi}_0 + \frac{1}{2}\phi_0 \otimes \pi_t$, where $\widetilde{\phi}_0 \in \mathscr{P}(\mathcal{S} \times \mathcal{A})$ is a base distribution and $\phi_0 = \int_{\mathcal{A}} \widetilde{\phi}_0(\cdot, \mathrm{d}a)$, the concentrability coefficient $\kappa$ is then given by

$$\kappa = \left\|\frac{\widetilde{\mathcal{E}}^{\pi^*}_{\mathcal{D}_0}}{\widetilde{\phi}_0}\right\|_\infty.$$

*Proof.* See §B.1 for a detailed proof. $\square$

The concentrability coefficient commonly appears in the reinforcement learning literature [7, 23, 24, 38, 39, 43, 48, 57, 61]. In contrast to a more standard concentrability coefficient form, note that $\kappa$ is irrelevant to the update of the algorithm. To show the convergence of the MF-PPO, our condition here is much weaker since we only need a given base distribution $\widetilde{\phi}_0$ such that $\kappa < \infty$.

Theorem 4.1 shows that the MF-PPO converges to the globally optimal policy at a rate of $\mathcal{O}(T^{-1})$ up to the policy evaluation error. Such a theorem implies the global optimality and convergence of a double-loop AC algorithm, where the critic $Q_t$ is solved to high precision and the policy evaluation error is sufficiently small. In the sequel, we consider a more challenging setting, where the critic $Q_t$ is updated simultaneously along with the update of the actor's policy $\pi_t$.

## 4.2 Global Optimality and Convergence of Two-timescale AC

In what follows, we aim to characterize the upper bound of the policy evaluation error when the critic and the actor are updated simultaneously. Specifically, the actor is updated via MF-PPO in (3.7) and the critic $Q_t = Q(\cdot; \rho_t)$ is updated via the MF-TD in (3.5). For the smooth function $\sigma$ in the parameterization of the Q function in (3.4), we consider it to be the following two-layer neural network,

$$\sigma(s, a; \theta) = B_\beta \cdot \beta(b) \cdot \widetilde{\sigma}(w^\top(s, a, 1)), \tag{4.2}$$

where $\widetilde{\sigma} : \mathbb{R} \to \mathbb{R}$ is the activation function, $\theta = (b, w)$ is the parameter, and $\beta : \mathbb{R} \to (-1, 1)$ is an odd and invertible function with scaling hyper-parameter $B_\beta > 0$. It then holds that $D = d + 2$, where $d$ and $D$ are the dimensions of $(s, a)$ and $\theta$, respectively. It is worth noting that the function class of $\int \sigma(s, a; \theta) \rho(\mathrm{d}\theta)$ for $\rho \in \mathscr{P}_2(\mathbb{R}^D)$ is the same as

$$\mathcal{F} = \left\{ \int \beta' \cdot \widetilde{\sigma}(w^\top(s, a, 1)) \nu(\mathrm{d}\beta', \mathrm{d}w) \,\middle|\, \nu \in \mathscr{P}_2((-B_\beta, B_\beta) \times \mathbb{R}^{d+1}) \right\}, \tag{4.3}$$

which captures a vast function class because of the universal function approximation theorem [8, 47]. We remark that we introduce the rescaling function $\beta$ in (4.2) to avoid the study of the space of probability measures over $(-B_\beta, B_\beta) \times \mathbb{R}^{d+1}$ in (4.3), which has boundary and thus lacks the regularity in the study of optimal transport. Furthermore, note that we introduce a hyper-parameter $\alpha > 1$ in the Q function in (3.4). Thus, we are using $\alpha \cdot \mathcal{F}$ to represent $\mathcal{F}$, which causes an "over-representation" when $\alpha > 1$. Such over-representation appears to be essential for our analysis. For a brief peek, we remark that $\alpha$ actually controls the gap in the average total reward over time when the relative time-scale $\eta$ is properly selected according to Theorem 4.6. Furthermore, such an influence is imposed through Lemma 4.4, which shows that the Wasserstein distance between $\rho_0$ and $\rho_{\pi_t}$ is upper bounded by $O(1/\alpha)$. In what follows, we consider the initialization of the TD update to be $\rho_0 = \mathcal{N}(0, I_D)$, which implies that $Q(s, a; \rho_0) = 0$. We next impose the following assumption on the two-layer neural network $\sigma$.

**Assumption 4.2** (Regularity of the Neural Network). For the two-layer neural network $\sigma$ defined in (4.2), we assume that the following properties hold.

(i) The rescaling function $\beta : \mathbb{R} \to (-1, 1)$ is odd, $L_{0,\beta}$-Lipschitz continuous, $L_{1,\beta}$-smooth, and invertible. Meanwhile, the inverse $\beta^{-1}$ is locally Lipschitz continuous. In particular, we assume that $\beta^{-1}$ is $\ell_\beta$-Lipschitz continuous in $[-2/3, 2/3]$.

(ii) The activation function $\widetilde{\sigma} : \mathbb{R} \to \mathbb{R}$ is odd, $B_{\widetilde{\sigma}}$-bounded, $L_{0,\widetilde{\sigma}}$-Lipschitz continuous, and $L_{1,\widetilde{\sigma}}$-smooth.

We remark that Assumption 4.2 is not restrictive and can be satisfied by a large family of neural networks, e.g., $\widetilde{\sigma}(x) = \tanh(x)$ and $\beta(b) = \tanh(b)$. Noting that $\|(s, a)\|_2 \leq 1$, Assumption 4.2 implies that the function $\sigma(s, a; \theta)$ in (4.2) is odd with respect to $w$ and $b$ and is also bounded, Lipschitz continuous, and smooth in the parameter domain, that is,

$$|\nabla_\theta \sigma(s, a; \theta)| < B_1, \quad |\nabla^2_{\theta\theta} \sigma(s, a; \theta)| < B_2. \tag{4.4}$$

We then impose the following assumption on the MDP.

**Assumption 4.3** (Regularity of the MDP). For the MDP $(\mathcal{S}, \mathcal{A}, \gamma, P, r, \mathcal{D}_0)$, we assume the following properties hold.

(i) The reward function $r$ and the transition kernel $P$ admit the following representations with respect to the activation function $\widetilde{\sigma}$,

$$r(s, a) = B_r \cdot \int \widetilde{\sigma}\big((s, a, 1)^\top w\big) \mu(\mathrm{d}w), \tag{4.5}$$

$$P(s' \,|\, s, a) = \int \widetilde{\sigma}\big((s, a, 1)^\top w\big) \varphi(s') \psi(s'; \mathrm{d}w), \tag{4.6}$$

where $\mu$ and $\psi(s'; \cdot)$ are probability measures in $\mathscr{P}_2(\mathbb{R}^{d+1})$ for any $s' \in \mathcal{S}$, $B_r$ is a positive scaling parameter, and $\varphi(s') : \mathcal{S} \to \mathbb{R}_+$ is a nonnegative function.

(ii) The reward function $r$ satisfies that $r(s, a) \geq 0$ for any $(s, a) \in \mathcal{S} \times \mathcal{A}$. For the representation of $r$ in (4.5) and the representation of the transition kernel $P$ in (4.6), we assume that

$$\chi^2(\mu \,\|\, \rho_{w,0}) < M_\mu, \qquad \chi^2\big(\psi(s; \cdot) \,\|\, \rho_{w,0}\big) < M_\psi, \quad \forall s \in \mathcal{S},$$

$$\int \varphi(s) \mathrm{d}s \leq M_{1,\varphi}, \qquad \int \varphi(s)^2 \mathrm{d}s \leq M_{2,\varphi},$$

where $\rho_{w,0}$ is the marginal distribution of $\rho_0$ with respect to $w$, i.e., $\rho_{w,0} = \int \rho_0(\mathrm{d}b, \cdot)$, $\chi^2$ is the chi-squared divergence, and $M_\mu$, $M_\psi$, $M_{1,\varphi}$, $M_{2,\varphi}$ are absolute constants.

(iii) We assume that there exists an absolute constant $\mathcal{G}$ such that

$$\big\|\psi(s; \cdot) - \psi(s'; \cdot)\big\|_{\dot{H}^{-1}(\mu)} < \mathcal{G}, \qquad \big\|\psi(s; \cdot) - \mu\big\|_{\dot{H}^{-1}(\mu)} < \mathcal{G},$$

$$\big\|\psi(s; \cdot) - \mu\big\|_{\dot{H}^{-1}(\psi(s'; \cdot))} < \mathcal{G}, \quad \big\|\psi(s; \cdot) - \psi(s'; \cdot)\big\|_{\dot{H}^{-1}(\psi(s''; \cdot))} < \mathcal{G}, \quad \forall s, s', s'' \in \mathcal{S},$$

where $\|\cdot\|_{\dot{H}^{-1}(\cdot)}$ is the weighted homogeneous Sobolev norm.

We remark that by assuming $\psi$ to be a probability measure and that $\varphi(s') \geq 0$ in (4.6), the representation of the transition kernel does not lose generality. Specifically, the function class of (4.6) is the same as

$$\mathcal{P} = \Big\{ \int \widetilde{\sigma}((s, a, 1)^\top w) \widetilde{\psi}(s'; \mathrm{d}w) \,\Big|\, \widetilde{\psi}(s'; \cdot) \text{ is a signed measure for any } s' \in \mathcal{S} \Big\}.$$

See §C.1 for a detailed proof. Assumption 4.3 generalizes the linear MDP in [14, 32, 67, 68]. In contrast, our representation of the reward function and the transition kernel benefits from the universal function approximation theorem and is thus not as restrictive as the original linear MDP assumption. Note that the infinite-width neural network has a two-layer structure by (4.2). We establish the following lemma on the regularity of the representation of the action value function $Q^\pi$ by such a neural network.

**Lemma 4.4** (Regularity of Representation of $Q^\pi$). Suppose that Assumptions 4.2 and 4.3 hold. For any policy $\pi$, there exists a probability measure $\rho_\pi \in \mathscr{P}_2(\mathbb{R}^D)$ for the representation of $Q^\pi$ with the following properties.

(i) For function $Q(s, a; \rho_\pi)$ defined by (3.4) with $\rho = \rho_\pi$ and the action value function $Q^\pi(s, a)$ defined by (2.2), we have $Q(s, a; \rho_\pi) = Q^\pi(s, a)$ for any $(s, a) \in \mathcal{S} \times \mathcal{A}$.

(ii) By letting $B_\beta \geq 2(B_r + \gamma(1 - \gamma)^{-1} B_r M_{1,\varphi})$ for the neural network defined in (4.2) and $\rho_0 \sim \mathcal{N}(0, I_D)$ for the initial distribution, we have $\widetilde{W}_2(\rho_\pi, \rho_0) \leq \bar{D}$ for any policy $\pi$, where we define $\widetilde{W}_2(\cdot, \cdot) = \alpha W_2(\cdot, \cdot)$ as the scaled $W_2$ metric. Here constant $\bar{D}$ depends on the discount factor $\gamma$ and the absolute constants $L_{0,\beta}, L_{1,\beta}, l_\beta, B_r, M_\mu, M_\psi, M_{1,\varphi}, M_{2,\varphi}$ defined in Assumptions 4.2 and 4.3.

*Proof.* See §B.2 for a detailed proof. □

Property (i) of Lemma 4.4 shows that the action value function $Q^\pi$ can be parameterized with the infinite-width two-layer neural network $Q(\cdot; \rho_\pi)$ in (3.4). Note that a larger $B_\beta$ captures a larger function class in (4.3). Without loss of generality, we consider that $B_\beta \geq 2(B_r + \gamma(1-\gamma)^{-1}B_r M_{1,\varphi})$ holds in the sequel. Hence, by Property (ii), it holds that $\widetilde{W}_2(\rho_\pi, \rho_0) \leq O(1)$ for any policy $\pi$. In particular, it holds by Property (i) of Lemma 4.4 that $\|Q_t - Q^{\pi_t}\|_{2,\widetilde{\phi}^{\pi_t}} = \|Q(\cdot; \rho_t) - Q(\cdot; \rho_{\pi_t})\|_{2,\widetilde{\phi}^{\pi_t}}$ and we have the following theorem to characterize such an error with regard to the $W_2$ space.

**Theorem 4.5** (Upper Bound of Policy Evaluation Error). Suppose that Assumptions 4.2 and 4.3 hold and $\rho_0 \sim \mathcal{N}(0, I_D)$ is the initial distribution. We specify the weighting distribution $\widetilde{\Phi}^{\pi_t}$ in MF-TD (3.5) as $\widetilde{\Phi}^{\pi_t} = \widetilde{\mathcal{E}}^{\pi_t}_{\widetilde{\phi}^{\pi_t}}$, where $\widetilde{\phi}^{\pi_t} \in \mathscr{P}(\mathcal{S} \times \mathcal{A})$ is the evaluation distribution for the policy evaluation error in Theorem 4.1. Then, it holds that

$$(1 - \sqrt{\gamma}) \cdot \|Q_t - Q^{\pi_t}\|^2_{2,\widetilde{\phi}^{\pi_t}} \leq -\frac{\mathrm{d}}{\mathrm{d}t} \frac{\widetilde{W}^2_2(\rho_t, \rho_{\pi_t})}{2\eta} + \Delta_t, \tag{4.7}$$

where

$$\begin{aligned}
\Delta_t =\, & 2\alpha^{1/2}\eta^{-1}\mathcal{B}B_1 \cdot \widetilde{W}_2(\rho_t, \rho_0)\widetilde{W}_2(\rho_t, \rho_{\pi_t}) \\
& + \alpha^{-1}B_2 \cdot \left(4B_1 \max\left\{\widetilde{W}_2(\rho_{\pi_t}, \rho_0), \widetilde{W}_2(\rho_t, \rho_0)\right\} + B_r\right)\widetilde{W}_2(\rho_t, \rho_\pi)^2.
\end{aligned}$$

Here $B_1$ and $B_2$ are defined in (4.4) of Assumption 4.2, $\eta$ is the relative TD timescale, $\alpha$ is the scaling parameter of the neural network, and $\widetilde{W}_2 = \alpha W_2$ is the scaled $W_2$ metric. Moreover, constant $\mathcal{B}$ depends on the dicount factor $\gamma$, the scaling parameter $B_\beta$ in (4.2), and the absolute constants $l_\beta, B_r, M_{1,\varphi}, \mathcal{G}$ defined in Assumptions 4.2 and 4.3.

*Proof.* See §B.3 for a detailed proof. $\qquad\square$

Here we give a nonrigorous discussion on how to upper bound $\Delta_t$ in (4.7). If $\widetilde{W}_2(\rho_t, \rho_0) \leq O(1)$ holds for any $t \in [0, T]$, by $\widetilde{W}_2(\rho_{\pi_t}, \rho_0) \leq O(1)$ in Lemma 4.4 and the triangle inequality of $W_2$ distance [59], it follows that $\widetilde{W}_2(\rho_t, \rho_{\pi_t}) \leq O(1)$ and $\Delta_t \leq O(\alpha^{1/2}\eta^{-1} + \alpha^{-1})$. Taking a time average of integration on both sides of (4.7), the policy evaluation error $\frac{1}{T}\int_0^T \|Q_t - Q^{\pi_t}\|_{2,\widetilde{\phi}^{\pi_t}} \mathrm{d}t$ is then upper bounded by $O(\eta^{-1}T^{-1} + \alpha^{1/2}\eta^{-1} + \alpha^{-1})$. Inspired by such a fact, we introduce the following restarting mechanism to ensure $\widetilde{W}_2(\rho_t, \rho_0) \leq O(1)$.

**Restarting Mechanism.** Let $\widetilde{W}_0 = \lambda\bar{D}$ be a threshold, where $\bar{D}$ is the upper bound for $\widetilde{W}_2(\rho_\pi, \rho_0)$ by Lemma 4.4, $\lambda \geq 3$ is a constant scaling parameter for the restarting threshold, $\rho_t$ is the distribution of the parameters in the neural network at time $t$, and $\rho_0$ is the initial distribution. Whenever we detect that $\widetilde{W}_2(\rho_t, \rho_0)$ reaches $\widetilde{W}_0$ in the update, we pause and reset $\rho_t$ to $\rho_0$ by resampling the parameters from $\rho_0$. Then, we reset the critic with the newly sampled parameters while keeping the actor's policy $\pi_t$ unchanged and continue the update.

The restarting mechanism guarantees $\widetilde{W}_2(\rho_t, \rho_0) \leq \lambda\bar{D}$ by restricting the distribution $\rho_t$ of the parameters to be close to $\rho_0$. Moreover, by letting $\lambda \geq 3$, we ensure that $\rho_{\pi_t}$ is realizable by $\rho_t$ since $\widetilde{W}_2(\rho_{\pi_t}, \rho_0) \leq \bar{D} \leq \lambda\bar{D}$, which means that the neural network is capable of capturing the representation of the action value function $Q^{\pi_t}$. We remark that by letting $\widetilde{W}_0 = O(1)$, we allow $\rho_t$ to deviate from $\rho_0$ up to $W_2(\rho_t, \rho_0) \leq O(\alpha^{-1})$ in the restarting mechanism. In contrast, the NTK regime [15] which corresponds to letting $\alpha = \sqrt{M}$ in (3.1) only allows $\rho_t$ to deviate from $\rho_0$ by the chi-squared divergence $\chi^2(\rho_t \| \rho_0) \leq O(M^{-1}) = o(1)$. That is, the NTK regime fails to induce a feature representation significantly different from the initial one. Before moving on, we summarize the construction of the weighting distribution $\widetilde{\Phi}^{\pi_t}$ in Theorem 4.1 and 4.5 as follows,

$$\widetilde{\Phi}^{\pi_t} = \widetilde{\mathcal{E}}^{\pi_t}_{\widetilde{\phi}^{\pi_t}}, \qquad \widetilde{\phi}^{\pi_t} = \frac{1}{2}\widetilde{\phi}_0 + \frac{1}{2}\phi_0 \otimes \pi_t, \qquad \phi_0 = \int_{\mathcal{A}} \widetilde{\phi}_0(\cdot, \mathrm{d}a), \tag{4.8}$$

where $\widetilde{\phi}_0$ is the base distribution. Now we have the following theorem that characterizes the global optimality and convergence of the two-timescale AC with restarting mechanism.

**Theorem 4.6** (Global Optimality and Convergence Rate of Two-timescale AC with Restarting Mechanism)**.** Suppose that (4.8) and Assumptions 4.2 and 4.3 hold. With the restarting mechanism, it holds that

$$\frac{1}{T}\int_0^T \big(J(\pi^*) - J(\pi_t)\big)\mathrm{d}t \leq \underbrace{\frac{\zeta}{T}}_{(a)} + \underbrace{4\kappa\sqrt{\alpha^{-1}S_1 + \alpha^{1/2}\eta^{-1}S_2 + \frac{\eta^{-1}\bar{D}^2}{2T(1-\sqrt{\gamma})}}}_{(b)}, \qquad (4.9)$$

where we have

$$\zeta = \mathbb{E}_{s\sim\mathcal{E}_{\mathcal{D}_0}^{\pi^*}}\Big[\mathrm{KL}\big(\pi^*(\cdot\,|\,s)\,\|\,\pi_0(\cdot\,|\,s)\big)\Big], \quad \kappa = \left\|\frac{\widetilde{\mathcal{E}}_{\mathcal{D}_0}^{\pi^*}}{\widetilde{\phi}_0}\right\|_\infty,$$

$$S_1 = \frac{(1+\lambda)^2\bar{D}^2 B_2(4B_1\lambda\bar{D} + B_r)}{1-\sqrt{\gamma}}, \quad S_2 = \frac{2\mathcal{B}B_1\lambda(1+\lambda)\bar{D}^2}{1-\sqrt{\gamma}}.$$

Here $B_r$, $B_1$ and $B_2$ are defined in Assumption 4.2 and 4.3, $\bar{D}$ is the upper bound for $\widetilde{W}_2(\rho_\pi, \rho_0)$ in Lemma 4.4, $\mathcal{B}$ depends on the discount factor $\gamma$ and the absolute constants defined in Assumption 4.2 and 4.3, and $\lambda$ is the scaling parameter for the restarting threshold. Besides, it holds for the total restarting number $N$ that

$$N \leq (\lambda - 2)^{-1}\big((\alpha^{-1}\eta S_1 + \alpha^{1/2}S_2)2T\bar{D}^{-2}(1-\sqrt{\gamma}) + 1\big).$$

*Proof.* See §B.4 for a detailed proof. □

Note that for a given MDP with starting distribution $\mathcal{D}_0$, the expected KL-divergence $\zeta$ and the concentrability coefficient $\kappa$ are both independent of the two-timescale update. We remark that our condition for (4.9) to be bounded is not restrictive. Specifically, we only need a given $\pi_0$ and $\widetilde{\phi}_0$ such that the KL-divergence $\zeta < \infty$ and the concentrability coefficient $\kappa < \infty$, which is weaker than the concentrability coefficient used in [7, 23, 24, 38, 39, 43, 48, 57, 61].

The first term (a) on the right-hand side of (4.9) diminishes as $T \to \infty$. The second term (b) corresponds to the policy evaluation error. We give an example to demonstrate the convergence of the two-time AC. We let the scaling parameter $\lambda = 3$ for the restarting threshold. By letting $\eta = \alpha^{3/2}$, it holds that (b) $= O(\alpha^{-1/2})$ as $\alpha \to \infty$. Thus, we have that $\frac{1}{T}\int_0^T \big(J(\pi^*) - J(\pi_t)\big)\mathrm{d}t$ descends at a rate of $O(T^{-1} + O(\alpha^{-1/2}) + O(\alpha^{-3/4}T^{-1/2}))$. Note that $\eta = \alpha^{3/2}$ shows that the critic has a larger relative TD timescale in (3.5). As for the total number of restartings $N$, it holds that $N \leq O(\alpha^{1/2}T)$ as $\alpha \to \infty$, which induces a tradeoff, i.e., a larger $\alpha$ guarantees a smaller gap in $\frac{1}{T}\int_0^T \big(J(\pi^*) - J(\pi_t)\big)\mathrm{d}t$ but yields in more restartings and a larger relative TD timescale.

## 5   Acknowledgement

Zhaoran Wang acknowledges National Science Foundation (Awards 2048075, 2008827, 2015568, 1934931), Simons Institute (Theory of Reinforcement Learning), Amazon, J.P. Morgan, and Two Sigma for their supports. Zhuoran Yang acknowledges Simons Institute (Theory of Reinforcement Learning). This work was also supported in part by the Vannevar Bush Faculty Fellowship program under grant number N00014-21-1-2941.

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
