# A    Supplement to the Background

In this section, we present some backgrounds on Wasserstein space and replicator dynamics.

## A.1    Wasserstein Space

Let $\Theta \subseteq \mathbb{R}^D$ be a Polish space. We denote by $\mathscr{P}_2(\Theta) \subseteq \mathscr{P}(\Theta)$ the set of probability measures with finite second moments. Then, the Wasserstein-2 ($W_2$) distance between $\mu, \nu \in \mathscr{P}_2(\Theta)$ is defined as follows,

$$W_2(\mu, \nu) = \inf\left\{\mathbb{E}\left[\|X - Y\|^2\right]^{1/2} \,\middle|\, \mathrm{law}(X) = \mu, \mathrm{law}(Y) = \nu\right\}, \tag{A.1}$$

where the infimum is taken over the random variables $X$ and $Y$ on $\Theta$ and we denote by $\mathrm{law}(X)$ the distribution of a random variable $X$. We call $\mathcal{M} = (\mathscr{P}_2(\Theta), W_2)$ the Wasserstein space, which is an infinite-dimensional manifold [59]. In particular, we define the tangent vector at $\mu \in \mathcal{M}$ as $\dot{\rho}_0$ for the corresponding curve $\rho : [0, 1] \to \mathscr{P}_2(\Theta)$ with $\rho_0 = \mu$. Under certain regularity conditions, the continuity equation $\partial_t \rho_t = -\operatorname{div}(\rho_t v_t)$ corresponds to a vector field $v : [0, 1] \times \Theta \to \mathbb{R}^D$, which endows the infinite-dimensional manifold $\mathscr{P}_2(\Theta)$ with a weak Riemannian structure in the following sense [59]. Given any tangent vectors $u$ and $\widetilde{u}$ at $\mu \in \mathcal{M}$ and the corresponding vector fields $v, \widetilde{v}$, which satisfy $u + \operatorname{div}(\mu v) = 0$ and $\widetilde{u} + \operatorname{div}(\mu \widetilde{v}) = 0$, respectively, we define the inner product of $u$ and $\widetilde{u}$ as follows,

$$\langle u, \widetilde{u} \rangle_{\mu, W_2} = \int v \cdot \widetilde{v} \mathrm{d}\mu = \langle v, \widetilde{v} \rangle_\mu, \tag{A.2}$$

which yields a Riemannian metric. Such a Riemannian metric further induces a norm $\|u\|_{\mu, W_2} = \langle u, u \rangle_{\mu, W_2}^{1/2}$ for any tangent vector $u \in T_\mu \mathcal{M}$ at any $\mu \in \mathcal{M}$, which allows us to write the Wasserstein-2 distance defined in (A.1) as follows,

$$W_2(\mu, \nu) = \inf\left\{\left(\int_0^1 \|\dot{\rho}_t\|_{\rho_t, W_2}^2 \, \mathrm{d}t\right)^{1/2} \,\middle|\, \rho : [0, 1] \to \mathcal{M}, \rho_0 = \mu, \rho_1 = \nu\right\}. \tag{A.3}$$

Here $\dot{\rho}_s$ denotes $\partial_t \rho_t \big|_{t=s}$ for any $s \in [0, 1]$. In particular, the infimum in (A.3) is attained by the geodesic $\widetilde{\rho} : [0, 1] \to \mathscr{P}_2(\Theta)$ connecting $\mu, \nu \in \mathcal{M}$. Moreover, the geodesics on $\mathcal{M}$ are constant-speed, that is,

$$\|\dot{\widetilde{\rho}}_t\|_{\widetilde{\rho}_t, W_2} = W_2(\mu, \nu), \quad \forall t \in [0, 1]. \tag{A.4}$$

In Wasserstein space $\mathcal{M}$, a curve $\rho : [0, 1] \to \mathscr{P}_2(\Theta)$ is defined to be absolutely continous if there exists $m \in L^1(a, b)$, i.e., $\int_a^b |\dot{m}(t)| \mathrm{d}t < \infty$, such that

$$W_2(\rho_s, \rho_t) \leq \int_s^t m(r) \mathrm{d}r, \quad \forall a < s \leq t < b.$$

Such an absolutely continuous curve $\rho_t$ allows us to define the metric derivative in $\mathcal{M}$ as follows,

$$|\dot{\rho}_t|_{W_2} = \lim_{s \to t} \frac{W_2(\rho_s, \rho_t)}{|s - t|}. \tag{A.5}$$

By [6], the metric derivative $|\dot{\rho}_t|_{W_2}$ is connected to the norm of the tangent vector by

$$|\dot{\rho}_t|_{W_2} = \|\dot{\rho}_t\|_{\rho_t, W_2}. \tag{A.6}$$

Furthermore, we introduce the Wasserstein-1 distance, which is defined as

$$W_1(\mu^1, \mu^2) = \inf\left\{\mathbb{E}\left[\|X - Y\|\right] \,\middle|\, \mathrm{law}(X) = \mu^1, \mathrm{law}(Y) = \mu^2\right\}$$

for any $\mu^1, \mu^2 \in \mathscr{P}(\mathbb{R}^D)$ with finite first moments. The Wasserstein-1 distance has the following dual representation [6],

$$W_1(\mu^1, \mu^2) = \sup\left\{\int f(x) \mathrm{d}(\mu^1 - \mu^2)(x) \,\middle|\, \text{continuous } f : \mathbb{R}^D \to \mathbb{R}, \mathrm{Lip}(f) \leq 1\right\}. \tag{A.7}$$

## A.2   Replicator Dynamics

The replicator dynamics originally arises in the study of evolutionary game theory [50]. For a function $f$, the replicator dynamics is given by the differential equation

$$\frac{\mathrm{d}}{\mathrm{d}t}x_t(a) = x_t(a)[f(a, x_t) - \phi(x)],$$

where $\phi(x) = \int x(a)f(a, x)$. As for the PPO update in (3.7), for a fixed $s$, let $x(a) = \pi(a \,|\, s)$ and $f(a, x) = Q^\pi(s, a)$, we see that (3.7) corresponds to a replicator dynamics if $Q_t = Q^{\pi_t}$. Note that in the simultaneous update of both the critic and actor, we do not have access to the true action value function $Q^\pi$. Thus, we use the estimator $Q_t$ calculated by the critic step to guide the update of the actor in the PPO update, which takes the form of a replicator dynamics in the continuous-time limit.

# B   Proofs of Main Results

In this section, we give detailed proof of the theorems and present a detailed statement of Lemma 4.4.

## B.1   Proof of Theorem 4.1

*Proof.* Following from the performance difference lemma [33], we have

$$J(\pi^*) - J(\pi_t) = (1-\gamma)^{-1} \cdot \mathbb{E}_{s \sim \mathcal{E}_{\mathcal{D}_0}^{\pi^*}}\big[\langle A^{\pi_t}(s, \cdot), \pi^*(\cdot \,|\, s) - \pi_t(\cdot \,|\, s)\rangle_{\mathcal{A}}\big],$$

where $\mathcal{E}_{\mathcal{D}_0}^{\pi^*}$ is the visitation measure induced by $\pi^*$ from $\mathcal{D}_0$ and $A^{\pi_t}$ is the advantage function. Note that the continuous PPO dynamics in (3.7) can be equivalently written as $\partial_t \log \pi_t = A_t$. Thus, we have

$$(1-\gamma) \cdot \big(J(\pi^*) - J(\pi_t)\big) = \mathbb{E}_{s \sim \mathcal{E}_{\mathcal{D}_0}^{\pi^*}}\Big[\big\langle \frac{\mathrm{d}}{\mathrm{d}t}\log \pi_t(\cdot \,|\, s) + A^{\pi_t}(s, \cdot) - A_t(s, \cdot), \pi^*(\cdot \,|\, s) - \pi_t(\cdot \,|\, s)\big\rangle_{\mathcal{A}}\Big]$$

$$= \mathbb{E}_{s \sim \mathcal{E}_{\mathcal{D}_0}^{\pi^*}}\Big[\big\langle \frac{\mathrm{d}}{\mathrm{d}t}\log \pi_t(\cdot \,|\, s), \pi^*(\cdot \,|\, s) - \pi_t(\cdot \,|\, s)\big\rangle_{\mathcal{A}}\Big]$$

$$+ \mathbb{E}_{s \sim \mathcal{E}_{\mathcal{D}_0}^{\pi^*}}\Big[\big\langle Q^{\pi_t}(s, \cdot) - Q_t(s, \cdot), \pi^*(\cdot \,|\, s) - \pi_t(\cdot \,|\, s)\big\rangle_{\mathcal{A}}\Big]. \qquad \text{(B.1)}$$

For the first term on the right-hand side of (B.1), it holds that,

$$\big\langle \frac{\mathrm{d}}{\mathrm{d}t}\log \pi_t(\cdot \,|\, s), \pi^*(\cdot \,|\, s) - \pi_t(\cdot \,|\, s)\big\rangle_{\mathcal{A}}$$

$$= \big\langle \frac{\mathrm{d}}{\mathrm{d}t}\log \pi_t(\cdot \,|\, s), \pi^*(\cdot \,|\, s)\big\rangle_{\mathcal{A}} - \big\langle \frac{\mathrm{d}}{\mathrm{d}t}\log \pi_t(\cdot \,|\, s), \pi_t(\cdot \,|\, s)\big\rangle_{\mathcal{A}} \qquad \text{(B.2)}$$

$$= -\frac{\mathrm{d}}{\mathrm{d}t}\mathrm{KL}\big(\pi^*(\cdot \,|\, s) \,\|\, \pi_t(\cdot \,|\, s)\big),$$

where the last equality holds by noting that $\langle \partial_t \log \pi_t(\cdot \,|\, s), \pi_t(\cdot \,|\, s)\rangle_{\mathcal{A}} = \partial_t \int_{\mathcal{A}} \pi_t(\mathrm{d}a \,|\, s) = 0$. For the second term on the right-hand side of (B.1), by the Cauchy-Schwartz inequality, we have

$$\mathbb{E}_{s \sim \mathcal{E}_{\mathcal{D}_0}^{\pi^*}}\Big[\big\langle Q^{\pi_t}(s, \cdot) - Q_t(s, \cdot), \pi^*(\cdot \,|\, s)\big\rangle_{\mathcal{A}}\Big] = \mathbb{E}_{(s,a) \sim \widetilde{\phi}^{\pi_t}}\Big[\big(Q^{\pi_t}(s, a) - Q_t(s, a)\big)\pi^*(a|s)\frac{\mathcal{E}_{\mathcal{D}_0}^{\pi^*}(s)}{\widetilde{\phi}^{\pi_t}(x)}\Big]$$

$$\leq \Big\|\frac{\widetilde{\mathcal{E}}_{\mathcal{D}_0}^{\pi^*}}{\widetilde{\phi}^{\pi_t}}\Big\|_{2,\widetilde{\phi}^{\pi_t}} \cdot \big\|Q^{\pi_t} - Q_t\big\|_{2,\widetilde{\phi}^{\pi_t}}, \qquad \text{(B.3)}$$

$$\mathbb{E}_{s \sim \mathcal{E}_{\mathcal{D}_0}^{\pi^*}}\Big[\big\langle Q^{\pi_t}(s, \cdot) - Q_t(s, \cdot), \pi_t(\cdot \,|\, s)\big\rangle_{\mathcal{A}}\Big] = \mathbb{E}_{(x) \sim \widetilde{\phi}^{\pi_t}}\Big[\big(Q^{\pi_t}(s, \cdot) - Q_t(s, \cdot)\big)\pi_t(a \,|\, s)\frac{\mathcal{E}_{\mathcal{D}_0}^{\pi^*}(s)}{\widetilde{\phi}^{\pi_t}(x)}\Big]$$

$$\leq \Big\|\frac{\mathcal{E}_{\mathcal{D}_0}^{\pi^*} \otimes \pi_t}{\widetilde{\phi}^{\pi_t}}\Big\|_{2,\widetilde{\phi}^{\pi_t}} \cdot \big\|Q^{\pi_t} - Q_t\big\|_{2,\widetilde{\phi}^{\pi_t}}. \qquad \text{(B.4)}$$

Plugging (B.2), (B.3), and (B.4) into (B.1), we have

$$J(\pi^*) - J(\pi_t) \leq -\frac{\mathrm{d}}{\mathrm{d}t}\mathbb{E}_{s\sim\mathcal{E}_{\mathcal{D}_0}^{\pi^*}}\Big[\mathrm{KL}\big(\pi^*(\cdot\,|\,s)\,\|\,\pi_t(\cdot\,|\,s)\big)\Big]$$
$$+ \Big(\Big\|\frac{\widetilde{\mathcal{E}}_{\mathcal{D}_0}^{\pi^*}}{\widetilde{\phi}^{\pi_t}}\Big\|_{2,\widetilde{\phi}^{\pi_t}} + \Big\|\frac{\mathcal{E}_{\mathcal{D}_0}^{\pi^*}\otimes\pi_t}{\widetilde{\phi}^{\pi_t}}\Big\|_{2,\widetilde{\phi}^{\pi_t}}\Big)\cdot\|Q_t - Q^{\pi_t}\|_{2,\widetilde{\phi}^{\pi_t}}. \qquad\text{(B.5)}$$

By further letting $\widetilde{\phi}^{\pi_t} = \frac{1}{2}\widetilde{\phi}_0 + \frac{1}{2}\phi_0\otimes\pi_t$, it holds for the concentrability coefficient $\Big\|\frac{\widetilde{\mathcal{E}}_{\mathcal{D}_0}^{\pi^*}}{\widetilde{\phi}^{\pi_t}}\Big\|_{2,\widetilde{\phi}^{\pi_t}}$ in (B.3) that

$$\Big\|\frac{\widetilde{\mathcal{E}}_{\mathcal{D}_0}^{\pi^*}}{\widetilde{\phi}^{\pi_t}}\Big\|_{2,\widetilde{\phi}^{\pi_t}} \leq \Big\|\frac{2\widetilde{\mathcal{E}}_{\mathcal{D}_0}^{\pi^*}}{\widetilde{\phi}_0}\Big\|_{2,\widetilde{\phi}^{\pi_t}} \leq 2\Big\|\frac{\widetilde{\mathcal{E}}_{\mathcal{D}_0}^{\pi^*}}{\widetilde{\phi}_0}\Big\|_\infty. \qquad\text{(B.6)}$$

By further letting $\phi_0(s) = \int_{\mathcal{A}}\widetilde{\phi}_0(s,\mathrm{d}a)$, it holds for the concentrability coefficient $\Big\|\frac{\mathcal{E}_{\mathcal{D}_0}^{\pi^*}\otimes\pi_t}{\widetilde{\phi}^{\pi_t}}\Big\|_{2,\widetilde{\phi}^{\pi_t}}$ in (B.4) that

$$\Big\|\frac{\mathcal{E}_{\mathcal{D}_0}^{\pi^*}\otimes\pi_t}{\widetilde{\phi}^{\pi_t}}\Big\|_{2,\widetilde{\phi}^{\pi_t}} \leq \Big\|\frac{2\mathcal{E}_{\mathcal{D}_0}^{\pi^*}}{\phi_0}\Big\|_{2,\widetilde{\phi}^{\pi_t}} \leq 2\Big\|\frac{\int\frac{\widetilde{\mathcal{E}}_{\mathcal{D}_0}^{\pi^*}(x)}{\widetilde{\phi}_0(x)}\widetilde{\phi}_0(s,\mathrm{d}a)}{\phi_0(s)}\Big\|_\infty \leq 2\Big\|\frac{\widetilde{\mathcal{E}}_{\mathcal{D}_0}^{\pi^*}}{\widetilde{\phi}_0}\Big\|_\infty. \qquad\text{(B.7)}$$

Plugging (B.6) and (B.7) into (B.5) and taking integration on both sides of (B.5), we have

$$\frac{1}{T}\int_0^T\big(J(\pi^*) - J(\pi_t)\big)\mathrm{d}t \leq \frac{1}{T}\mathbb{E}_{s\sim\mathcal{E}_{\mathcal{D}_0}^{\pi^*}}\Big[\mathrm{KL}\big(\pi^*(\cdot\,|\,s)\,\|\,\pi_0(\cdot\,|\,s)\big)\Big] + \frac{4\kappa}{T}\cdot\int_0^T\|Q_t - Q^{\pi_t}\|_{2,\widetilde{\phi}^{\pi_t}}\mathrm{d}t,$$

where $\kappa = \big\|\widetilde{\mathcal{E}}_{\mathcal{D}_0}^{\pi^*}/\widetilde{\phi}_0\big\|_\infty$. Thus, we complete the proof of Theorem 4.1. □

## B.2 Detailed Statement of Lemma 4.4

We give a detailed version of Lemma 4.4 as follows.

**Lemma B.1** (Regularity of Representation of $Q^\pi$). Suppose that Assumptions 4.2 and 4.3 hold. For any policy $\pi$, there exists a probability measure $\rho_\pi \in \mathscr{P}_2(\mathbb{R}^D)$ for the representation of $Q^\pi$ with the following properties.

(i) For function $Q(s,a;\rho_\pi)$ defined by (3.4) with $\rho = \rho_\pi$ and the action value function $Q^\pi(s,a)$ defined by (2.2), we have $Q(s,a;\rho_\pi) = Q^\pi(s,a)$ for any $(s,a)\in\mathcal{S}\times\mathcal{A}$.

(ii) For $g$ defined in (3.6), we have $g(\cdot;\rho_\pi) = 0$ for any policy $\pi$.

(iii) By letting $B_\beta \geq 2(B_r + \gamma(1-\gamma)^{-1}B_r M_{1,\varphi})$ for the neural network defined in (4.2) and $\rho_0 \sim \mathcal{N}(0, I_D)$ for the initial distribution, we have $\widetilde{W}_2(\rho_\pi, \rho_0) < \bar{D}$ for any policy $\pi$, where we define $\widetilde{W}(\cdot,\cdot) = \alpha W(\cdot,\cdot)$ as the scaled $W_2$ metric. Here constant $\bar{D}$ depends on the discount factor $\gamma$ and the absolute constants $L_{0,\beta}, L_{1,\beta}, l_\beta, B_r, M_\mu, M_\psi, M_{1,\varphi}, M_{2,\varphi}$ defined in Assumptions 4.2 and 4.3.

(iv) For any two policies $\pi_1$ and $\pi_2$, it holds that,

$$W_2(\rho_{\pi_1}, \rho_{\pi_2}) \leq \alpha^{-\frac{1}{2}}\mathcal{B}\cdot\sup_{s\in\mathcal{S}}\mathbb{E}_{s'\sim\mathcal{E}_s^{\pi_1}}\Big[\big\|\pi_1(\cdot\,|\,s') - \pi_2(\cdot\,|\,s')\big\|_1\Big],$$

where $\mathcal{B}$ depends on the dicount factor $\gamma$, the scaling parameter $B_\beta$ in (4.2), and the absolute constants $l_\beta, B_r, M_{1,\varphi}, \mathcal{G}$ defined in Assumptions 4.2 and 4.3.

*Proof.* See §C.2 for a detailed proof. □

## B.3 Proof of Theorem 4.5

*Proof.* For notation simplicity, we let $x = (s, a)$. By Property (i) of Lemma B.1, it holds that $\|Q_t - Q^{\pi_t}\|^2_{2, \widetilde{\phi}^{\pi_t}} = \|Q_t - Q(\cdot; \rho_\pi)\|^2_{2, \widetilde{\phi}^{\pi_t}}$, where $Q_t = Q(\cdot; \rho_t)$. Thus it prompts us to study the $W_2$ distance between $\rho_t$ and $\rho_{\pi_t}$. By the first variation formula in Lemma E.2, it holds that

$$\frac{\mathrm{d}}{\mathrm{d}t} \frac{W_2^2(\rho_t, \rho_{\pi_t})}{2} = -\langle \dot{\rho}_t, \dot{\widetilde{\alpha}}_t^0 \rangle_{\rho_t, W_2} - \langle \dot{\rho}_{\pi_t}, \dot{\widetilde{\beta}}_t^0 \rangle_{\rho_{\pi_t}, W_2}. \tag{B.8}$$

Here $\widetilde{\alpha}_t^{[0,1]}$ is the geodesic connecting $\rho_t$ and $\rho_{\pi_t}$, and $\widetilde{\beta}_t^{[0,1]}$ is its time-inverse, i.e., $\widetilde{\beta}_t^s = \widetilde{\alpha}_t^{1-s}$. Besides, we denote by $\dot{\widetilde{\alpha}}_t^s = \partial_s \widetilde{\alpha}_t^s$ and $\dot{\widetilde{\beta}}_t^s = \partial_s \widetilde{\beta}_t^s$ the derivation of geodesics $\widetilde{\alpha}_t^s$ and $\widetilde{\beta}_t^s$ with respect to $s$, respectively. We denote by $v_s$ the corresponding vector field at $\widetilde{\alpha}_t^s$, which satisfies $\partial_s \widetilde{\alpha}_t^s = -\mathrm{div}(\widetilde{\alpha}_t^s v_s)$. For the first term of (B.8), it holds that

$$-\langle \dot{\rho}_t, \dot{\widetilde{\alpha}}_t^0 \rangle_{\rho_t, W_2} = -\eta \cdot \langle g(\cdot; \rho_t, \pi_t), v_0 \rangle_{\rho_t}$$

$$= \eta \cdot \int_0^1 \partial_s \langle g(\cdot; \widetilde{\alpha}_t^s, \pi_t), v_s \rangle_{\widetilde{\alpha}_t^s} \mathrm{d}s - \eta \cdot \langle g(\cdot; \rho_{\pi_t}, \pi_t), v_1 \rangle_{\rho_{\pi_t}}$$

$$= \eta \cdot \int_0^1 \langle \partial_s g(\cdot; \widetilde{\alpha}_t^s, \pi_t), v_s \rangle_{\widetilde{\alpha}_t^s} \mathrm{d}s + \eta \cdot \int_0^1 \int g(\theta; \widetilde{\alpha}_t^s, \pi_t) \cdot \partial_s(v_s \widetilde{\alpha}_t^s)(\theta) \mathrm{d}\theta \mathrm{d}s, \tag{B.9}$$

where the first equality follows from (A.2) and the third equation follows from $g(\cdot; \rho_{\pi_t}, \pi_t) = 0$ by Property (ii) of Lemma B.1. For the first term on the right-hand side of (B.9), we have

$$\eta \cdot \int_0^1 \langle \partial_s g(\cdot; \widetilde{\alpha}_t^s, \pi_t), v_s \rangle_{\widetilde{\alpha}_t^s} \mathrm{d}s$$

$$= -\alpha^{-1} \eta \cdot \int_0^1 \int \left\langle \mathbb{E}_{\widetilde{\Phi}^{\pi_t}}^{\pi_t} \left[ \partial_s \left( Q(x; \widetilde{\alpha}_t^s) - \gamma \cdot Q(x'; \widetilde{\alpha}_t^s) \right) \nabla \sigma(x; \theta) \right], (v_s \widetilde{\alpha}_t^s)(\theta) \right\rangle \mathrm{d}\theta \mathrm{d}s$$

$$= \alpha^{-1} \eta \cdot \int_0^1 \int \left\langle \mathbb{E}_{\widetilde{\Phi}^{\pi_t}}^{\pi_t} \left[ \partial_s \left( Q(x; \widetilde{\alpha}_t^s) - \gamma \cdot Q(x'; \widetilde{\alpha}_t^s) \right) \sigma(x; \theta) \right], \mathrm{div}(v_s \widetilde{\alpha}_t^s)(\theta) \right\rangle \mathrm{d}\theta \mathrm{d}s, \tag{B.10}$$

where the first equality holds by defintion of $g$ in (3.6) and the last equality follows from Stokes' formula. Note that we have $\mathrm{div}(v_s \widetilde{\alpha}_t^s) = -\partial_s \widetilde{\alpha}_t^s$ by the definition of vector field $v_s$. Thus, it holds for (B.10) that

$$\eta \cdot \int_0^1 \langle \partial_s g(\cdot; \widetilde{\alpha}_t^s, \pi_t), v_s \rangle_{\widetilde{\alpha}_t^s} \mathrm{d}s$$

$$= -\alpha^{-1} \eta \cdot \int_0^1 \int \left\langle \mathbb{E}_{\widetilde{\Phi}^{\pi_t}}^{\pi_t} \left[ \partial_s \left( Q(x; \widetilde{\alpha}_t^s) - \gamma \cdot Q(x'; \widetilde{\alpha}_t^s) \right) \sigma(x; \theta) \right], \partial_s \widetilde{\alpha}_t^s(\theta) \right\rangle \mathrm{d}\theta \mathrm{d}s$$

$$= -\alpha^{-2} \eta \cdot \int_0^1 \mathbb{E}_{\widetilde{\Phi}^{\pi_t}}^{\pi_t} \left[ \partial_s \left( Q(x; \widetilde{\alpha}_t^s) - \gamma \cdot Q(x'; \widetilde{\alpha}_t^s) \right) \partial_s Q(x; \widetilde{\alpha}_t^s) \right] \mathrm{d}s, \tag{B.11}$$

where the last equality follows from the definition of $Q$ in (3.4). We let $f(\widetilde{D}) = \left( \mathbb{E}_{x \sim \widetilde{\mathcal{E}}_{\widetilde{D}}^{\pi_t}} [(\partial_s Q(x; \widetilde{\alpha}_t^s))^2] \right)^{1/2}$ with respect to a specific $s$ and $t$. Recall that for the weighting distribution $\widetilde{\Phi}^{\pi_t}$, we set $\widetilde{\Phi}^{\pi_t} = \widetilde{\mathcal{E}}_{\widetilde{\phi}^{\pi_t}}^{\pi_t}$. Hence, for the integrand of (B.11), we have

$$-\mathbb{E}_{\widetilde{\Phi}^{\pi_t}}^{\pi_t} \left[ \partial_s \left( Q(x; \widetilde{\alpha}_t^s) - \gamma \cdot Q(x'; \widetilde{\alpha}_t^s) \right) \partial_s Q(x; \widetilde{\alpha}_t^s) \right]$$

$$= -f(\widetilde{\phi}^{\pi_t})^2 + \gamma \cdot \int \partial_s Q(x; \widetilde{\alpha}_t^s) \partial_s Q(x'; \widetilde{\alpha}_t^s) \widetilde{P}^{\pi_t}(x' \mid x) \widetilde{\mathcal{E}}_{\widetilde{\phi}^{\pi_t}}^{\pi_t}(x) \mathrm{d}x' \mathrm{d}x$$

$$\leq -f(\widetilde{\phi}^{\pi_t})^2 + \gamma \cdot \sqrt{\int \left( \partial_s Q(x; \widetilde{\alpha}_t^s) \right)^2 \widetilde{\mathcal{E}}_{\widetilde{\phi}^{\pi_t}}^{\pi_t}(\mathrm{d}x)} \cdot \sqrt{\int \left( \partial_s Q(x'; \widetilde{\alpha}_t^s) \right)^2 \widetilde{P}^{\pi_t}(\mathrm{d}x' \mid x) \widetilde{\mathcal{E}}_{\widetilde{\phi}^{\pi_t}}^{\pi_t}(\mathrm{d}x)}, \tag{B.12}$$

where the equality follows from the definition of $\mathbb{E}^{\pi_t}_{\widetilde{\Phi}^{\pi_t}}$ in (3.6) and the inequality folllows from the Cauchy-Schwarz inequality. We define $\mathcal{T}^\pi : \mathscr{P}(\mathcal{S} \times \mathcal{A}) \to \mathscr{P}(\mathcal{S} \times \mathcal{A})$ as a mapping operator such that $\mathcal{T}^\pi \widetilde{\mathcal{D}}(x') = \int \widetilde{\mathcal{D}}(x) \widetilde{P}^\pi(x'|\mathrm{d}x)$. We rewrite (B.12) as

$$-\mathbb{E}^{\pi_t}_{\widetilde{\Phi}^{\pi_t}}\big[\partial_s\big(Q(x;\widetilde{\alpha}^s_t) - \gamma \cdot Q(x';\widetilde{\alpha}^s_t)\big)\partial_s Q(x;\widetilde{\alpha}^s_t)\big] \leq -f(\widetilde{\phi}^{\pi_t})^2 + \gamma \cdot f(\widetilde{\phi}^{\pi_t}) \cdot f(\mathcal{T}^{\pi_t}\widetilde{\phi}^{\pi_t}).$$
(B.13)

By the definition of $\mathcal{T}^{\pi_t}$ and the definition of visitation measure in (2.1), it holds that $\widetilde{\mathcal{E}}^{\pi_t}_{\widetilde{\phi}^{\pi_t}} - \gamma \widetilde{\mathcal{E}}^{\pi_t}_{\mathcal{T}^{\pi_t}\widetilde{\phi}^{\pi_t}} = (1-\gamma)\widetilde{\phi}^{\pi_t}$. Hence, we have $f(\widetilde{\phi}^{\pi_t})^2 - \gamma f^2(\mathcal{T}^{\pi_t}\widetilde{\phi}^{\pi_t}) = (1-\gamma)\mathbb{E}_{\widetilde{\phi}^{\pi_t}}\big[\big(\partial_s Q(x;\widetilde{\alpha}^s_t)\big)^2\big]$ and it holds for (B.13) that

$$
\begin{aligned}
-f(\widetilde{\phi}^{\pi_t})^2 + \gamma f(\widetilde{\phi}^{\pi_t}) \cdot f(\mathcal{T}^{\pi_t}\widetilde{\phi}^{\pi_t}) &= -\frac{f(\widetilde{\phi}^{\pi_t})}{f(\widetilde{\phi}^{\pi_t}) + \gamma f(\mathcal{T}^{\pi_t}\widetilde{\phi}^{\pi_t})} \cdot \big(f(\widetilde{\phi}^{\pi_t})^2 - \gamma^2 f^2(\mathcal{T}^{\pi_t}\widetilde{\phi}^{\pi_t})\big) \\
&\leq -\frac{f(\widetilde{\phi}^{\pi_t})^2 - \gamma\Big(f(\widetilde{\phi}^{\pi_t})^2 - (1-\gamma)\mathbb{E}_{\widetilde{\phi}^{\pi_t}}\big[\big(Q(x;\widetilde{\alpha}^s_t)\big)^2\big]\Big)}{1 + \sqrt{\gamma}} \\
&\leq -(1-\sqrt{\gamma}) \cdot \mathbb{E}_{\widetilde{\phi}^{\pi_t}}\big[\big(\partial_s Q(x;\widetilde{\alpha}^s_t)\big)^2\big],
\end{aligned}
$$
(B.14)

where the first inequality holds by noting that $f(\widetilde{\phi}^{\pi_t}) \geq \sqrt{\gamma}f(\mathcal{T}^{\pi_t}\widetilde{\phi}^{\pi_t})$ and the last inequality holds by noting that $f(\widetilde{\phi}^{\pi_t})^2 \geq (1-\gamma)\mathbb{E}_{\widetilde{\phi}^{\pi_t}}\big[\big(\partial_s Q(x;\widetilde{\alpha}^s_t)\big)^2\big]$. Combining (B.11), (B.13), and (B.14) together, it holds for the first term of (B.9) that

$$
\begin{aligned}
\eta \cdot \int_0^1 \langle \partial_s g(\cdot;\widetilde{\alpha}^s_t,\pi_t), v_s\rangle_{\widetilde{\alpha}^s_t}\mathrm{d}s &\leq -\alpha^{-2}\eta(1-\sqrt{\gamma}) \cdot \int_0^1 \mathbb{E}_{\widetilde{\phi}^{\pi_t}}\big[\big(\partial_s Q(x;\widetilde{\alpha}^s_t)\big)^2\big]\mathrm{d}s \\
&\leq -\alpha^{-2}\eta(1-\sqrt{\gamma}) \cdot \big\|Q(x;\rho_{\pi_t}) - Q(x;\rho_t)\big\|^2_{2,\widetilde{\phi}^{\pi_t}},
\end{aligned}
$$
(B.15)

where the last inequality holds by the Cauchy-Schwarz inequality. For the second term of (B.9), we have

$$
\begin{aligned}
\eta \cdot \int_0^1 \int g(\theta;\widetilde{\alpha}^s_t,\pi_t) \cdot \partial_s(v_s \widetilde{\alpha}^s_t)(\theta)\mathrm{d}\theta\mathrm{d}s &= \eta \cdot \int_0^1 \int \langle \nabla g(\theta;\widetilde{\alpha}^s_t,\pi_t), \widetilde{\alpha}^s_t(\theta) \cdot v_s(\theta) \otimes v_s(\theta)\rangle\mathrm{d}\theta\mathrm{d}s \\
&\leq \eta \cdot \int_0^1 \sup_\theta\|\nabla g(\theta;\widetilde{\alpha}^s_t,\pi_t)\|_{\mathrm{F}} \cdot W_2(\rho_t,\rho_{\pi_t})^2\mathrm{d}s \\
&= \eta \cdot \sup_{\theta,s}\|\nabla g(\theta;\widetilde{\alpha}^s_t,\pi_t)\|_{\mathrm{F}} \cdot W_2(\rho_t,\rho_{\pi_t})^2,
\end{aligned}
$$
(B.16)

where the first euality holds by Eulerian representation of geodesic in Lemma E.3 that $\partial_s(v_s \cdot \widetilde{\alpha}^s_t) = -\operatorname{div}(\widetilde{\alpha}^s_t \cdot v_s \otimes v_s)$ and Stokes's fomula. Here, we denote by $\otimes$ the outer product between two vectors. The inequlity of (B.16) follows from $\|v_s(\theta) \otimes v_s(\theta)\|_{\mathrm{F}} = \|v_s(\theta)\|^2$ and the property of geodesic in (A.4) that $\|v_s\|_{\widetilde{\alpha}^s_t,W_2} = W_2(\rho_t,\rho_{\pi_t})$.

For the Second term on the right-hand side of (B.8), we denote by $u_t$ the corresponding vector field at $\rho_{\pi_t}$ such that $\partial_t \rho_{\pi_t} = -\operatorname{div}(\rho_{\pi_t} u_t)$. Then, it holds that

$$-\langle \dot{\rho}_{\pi_t}, \dot{\widetilde{\beta}}^0_t\rangle_{\rho_{\pi_t},W_2} = \langle u_t, v_1\rangle_{\rho_{\pi_t}} \leq \|\dot{\rho}_{\pi_t}\|_{\rho_{\pi_t},W_2} \cdot W_2(\rho_t,\rho_{\pi_t}) = |\dot{\rho}_{\pi_t}|_{W_2} \cdot W_2(\rho_t,\rho_{\pi_t}),$$
(B.17)

where the first equality follows from (A.2), the inequality follows from the Cauchy-Schwarz inequality and the facts that $\|u_t\|_{\rho_{\pi_t}} = \|\dot{\rho}_{\pi_t}\|_{\rho_{\pi_t},W_2}$ by (A.2) and that $\|v_1\|_{\rho_{\pi_t}} = W_2(\rho_t,\rho_{\pi_t})$ by (A.4). The last equality of (B.17) holds by (A.6). Plugging the definition of metric derivative $|\dot{\rho}_t|_{W_2}$ in (A.5) into (B.17), we have

$$
\begin{aligned}
-\langle \dot{\rho}_{\pi_t}, \dot{\widetilde{\beta}}^0_t\rangle_{\rho_{\pi_t},W_2} &\leq \lim_{\Delta t \to 0}\frac{W_2(\rho_{\pi_t},\rho_{\pi_{t+\Delta t}})}{|\Delta t|} \cdot W_2(\rho_t,\rho_{\pi_t}) \\
&\leq \alpha^{-1/2}\mathcal{B} \cdot \lim_{\Delta t \to 0}\sup_{s \in \mathcal{S}}\mathbb{E}_{s' \sim \mathcal{E}^{\pi_t}_s}\Big[\Big\|\frac{\pi_{t+\Delta t}(\cdot\,|\,s') - \pi_t(\cdot\,|\,s')}{\Delta t}\Big\|_1\Big] \cdot W_2(\rho_t,\rho_{\pi_t}) \\
&= \alpha^{-1/2}\mathcal{B} \cdot \sup_{s \in \mathcal{S}}\mathbb{E}_{s' \sim \mathcal{E}^{\pi_t}_s}\big[\big\|A_t(s',\cdot)\pi_t(\cdot\,|\,s')\big\|_1\big] \cdot W_2(\rho_t,\rho_{\pi_t}),
\end{aligned}
$$
(B.18)

where the second inequality follows from Property (iv) of Lemma B.1 and the equality follows from the MF-PPO update in (3.7). For the approximation of the advantage function $A_t$, it holds that

$$\sup_{x \in \mathcal{S} \times \mathcal{A}} |A_t(x)| = \sup_{x \in \mathcal{S} \times \mathcal{A}} \left| Q(x; \rho_t) - \int Q(s, a') \pi_t(\mathrm{d}a' \,|\, s) \right|$$
$$\le 2 \sup_{x \in \mathcal{S} \times \mathcal{A}} |Q(x; \rho_t)|. \tag{B.19}$$

Plugging (B.19) into (B.18), we have

$$-\langle \dot{\rho}_{\pi_t}, \widetilde{\beta}_t^0 \rangle_{\rho_{\pi_t}} \le \alpha^{-1/2} \mathcal{B} \cdot \sup_{x \in \mathcal{X}} |A_t(x)| \cdot \sup_{s \in \mathcal{S}} \mathbb{E}_{s' \sim \mathcal{E}_s^{\pi_t}} \left[ \|\pi_t(\cdot \,|\, s')\|_1 \right] \cdot W_2(\rho_t, \rho_{\pi_t})$$
$$\le 2\alpha^{-1/2} \mathcal{B} \cdot \sup_{x \in \mathcal{X}} |Q(x; \rho_t)| \cdot W_2(\rho_t, \rho_{\pi_t}). \tag{B.20}$$

where the last inequality follows from $\|\pi_t(\cdot \,|\, s')\|_1 = 1$. By first plugging (B.15) and (B.16) into (B.9), and then plugging (B.9) and (B.20) into (B.8), we have

$$\frac{\mathrm{d}}{\mathrm{d}t} \frac{W_2^2(\rho_t, \rho_{\pi_t})}{2} \le -\alpha^{-2} \eta(1 - \sqrt{\gamma}) \cdot \|Q_t - Q^{\pi_t}\|_{2, \widetilde{\phi}^{\pi_t}}^2 + \eta \cdot \sup_{\theta, s} \|\nabla g(\theta; \widetilde{\alpha}_t^s, \pi_t)\|_{\mathrm{F}} \cdot W_2(\rho_t, \rho_\pi)^2$$
$$+ 2\alpha^{-1/2} \mathcal{B} \cdot \sup_{x \in \mathcal{X}} |Q(x; \rho_t)| \cdot W_2(\rho_t, \rho_{\pi_t}). \tag{B.21}$$

Plugging Lemma D.1 into (B.21), we have

$$\frac{\mathrm{d}}{\mathrm{d}t} \frac{W_2^2(\rho_t, \rho_{\pi_t})}{2} \le - \eta \alpha^{-2} (1 - \sqrt{\gamma}) \cdot \|Q_t - Q^{\pi_t}\|_{2, \widetilde{\phi}^{\pi_t}}^2$$
$$+ \eta \alpha^{-1} B_2 \cdot \left( 2\alpha B_1 \sup_{s \in [0,1]} W_2(\widetilde{\alpha}_t^s, \rho_0) + B_r \right) \cdot W_2(\rho_t, \rho_\pi)^2$$
$$+ 2\alpha^{1/2} \mathcal{B} B_1 \cdot W_2(\rho_t, \rho_0) W_2(\rho_t, \rho_{\pi_t}). \tag{B.22}$$

Note that $\widetilde{\alpha}_t^s$ is the geodesic connecting $\rho_t$ and $\rho_\pi$. By Lemma D.2, we have

$$\sup_{s \in [0,1]} W_2(\widetilde{\alpha}_t^s, \rho_0) \le 2 \max \left\{ W_2(\rho_{\pi_t}, \rho_0), W_2(\rho_t, \rho_0) \right\}. \tag{B.23}$$

Plugging (B.23) into (B.22), it follows that

$$\frac{\mathrm{d}}{\mathrm{d}t} \frac{\widetilde{W}_2^2(\rho_t, \rho_{\pi_t})}{2\eta} \le - (1 - \sqrt{\gamma}) \cdot \|Q_t - Q^{\pi_t}\|_{2, \widetilde{\phi}^{\pi_t}}^2 + 2\eta^{-1} \alpha^{1/2} \mathcal{B} B_1 \cdot \widetilde{W}_2(\rho_t, \rho_0) \widetilde{W}_2(\rho_t, \rho_{\pi_t})$$
$$+ \alpha^{-1} B_2 \cdot \left( 4B_1 \max \left\{ \widetilde{W}_2(\rho_{\pi_t}, \rho_0), \widetilde{W}_2(\rho_t, \rho_0) \right\} + B_r \right) \widetilde{W}_2(\rho_t, \rho_\pi)^2,$$

Where $\widetilde{W}_2 = \alpha^{-1} W_2$ is the scaled $W_2$ metric. Thus, we complete the proof of Theorem 4.5. $\qquad \square$

## B.4  Proof of Theorem 4.6

*Proof.* We remark that the restarting mechanism produces discontinuity in $\rho_t$ while $\pi_t$ remains continuous. Let $T_0, T_1, \cdots, T_N$ denote the restarting points in $[0, T)$, where $T_0 = 0$ and $N$ is the total restarting number in $[0, T)$. Let $T_n^-$ and $T_n^+$ denote the moments just before and after the restarting occurring at $T_n$, respectively. According to the restarting mechanism, we have $\widetilde{W}_2(\rho_t, \rho_0) \le \lambda \bar{D}$, $\widetilde{W}_2(\rho_{T_n^-}, \rho_0) = \lambda \bar{D}$ and $\rho_{T_n^+} = \rho_0$. Recall that we set $\widetilde{\Phi}^{\pi_t} = \widetilde{\mathcal{E}}_{\widetilde{\phi}^{\pi_t}}^{\pi_t}$. By (4.7) of Theorem 4.5, it holds that

$$(1 - \sqrt{\gamma}) \cdot \|Q_t - Q^{\pi_t}\|_{2, \widetilde{\phi}^{\pi_t}}^2 \le - \frac{\mathrm{d}}{\mathrm{d}t} \frac{\widetilde{W}_2^2(\rho_t, \rho_{\pi_t})}{2\eta} + 2\eta^{-1} \alpha^{1/2} \mathcal{B} B_1 \lambda \bar{D} \cdot \widetilde{W}_2(\rho_t, \rho_{\pi_t})$$
$$+ \alpha^{-1} B_2 \cdot (4B_1 \lambda \bar{D} + B_r) \widetilde{W}_2(\rho_t, \rho_\pi)^2. \tag{B.24}$$

For simplicity, we let $S_1 = \frac{(1+\lambda)^2 \bar{D}^2 B_2 (4B_1 \lambda \bar{D} + B_r)}{1 - \sqrt{\gamma}}$, $S_2 = \frac{2 \mathcal{B} B_1 \lambda (1+\lambda) \bar{D}^2}{1 - \sqrt{\gamma}}$, $\xi = \frac{1}{2(1 - \sqrt{\gamma})}$, $Q(t) = \|Q_t - Q^{\pi_t}\|_{2, \widetilde{\phi}^{\pi_t}}$, and $\widetilde{W}(t) = \widetilde{W}_2(\rho_t, \rho_{\pi_t})$. By sum of the integrals of (B.24) on

$[T_0, T_1], \cdots, [T_{N-1}, T_N]$, and $[T_N, T]$, we have

$$\frac{1}{T}\int_0^T Q^2(t)\mathrm{d}t \le \frac{1}{T}\int_0^T \Big(\frac{\alpha^{-1}S_1}{(1+\lambda)^2\bar{D}^2}\widetilde{W}(t)^2 + \frac{\alpha^{1/2}\eta^{-1}S_2}{(1+\lambda)\bar{D}}\widetilde{W}(t)\Big)\mathrm{d}t$$
$$-\frac{\xi}{T\eta}\Big(\sum_{n=0}^{N-1}\int_{T_n}^{T_{n+1}}\frac{\mathrm{d}}{\mathrm{d}t}\widetilde{W}(t)^2\mathrm{d}t + \int_{T_N}^T\frac{\mathrm{d}}{\mathrm{d}t}\widetilde{W}(t)^2\mathrm{d}t\Big). \qquad \text{(B.25)}$$

Note that we have $\widetilde{W}(t) = \widetilde{W}_2(\rho_t, \rho_{\pi_t}) \le \widetilde{W}_2(\rho_t, \rho_0) + \widetilde{W}_2(\rho_{\pi_t}, \rho_0) \le (1+\lambda)\bar{D}$ by the triangle inequality of $W_2$ distance, the resarting mechanism and Property (iii) of Lemma B.1. It thus holds for (B.25) that

$$\frac{1}{T}\int_0^T Q^2(t)\mathrm{d}t \le \frac{1}{T}\int_0^T (\alpha^{-1}S_1 + \alpha^{1/2}\eta^{-1}S_2)\mathrm{d}t$$
$$-\frac{\xi}{T\eta}\Big(\sum_{n=0}^{N-1}\big(\widetilde{W}^2(T_{n+1}^-) - \widetilde{W}^2(T_n^+)\big) + \big(\widetilde{W}^2(T) - \widetilde{W}^2(T_N^+)\big)\Big). \qquad \text{(B.26)}$$

Note that we have $\widetilde{W}(T_{n+1}^-) \ge \widetilde{W}_2(\rho_0, \rho_{T_{n+1}^-}) - \widetilde{W}_2(\rho_0, \rho_{\pi_{T_{n+1}^-}}) \ge (\lambda - 1)\bar{D} \ge \bar{D} \ge \widetilde{W}(\rho_0, \rho_{\pi_{T_n^+}}) = \widetilde{W}(T_n^+)$ by the triangle inequality of $W_2$ distance, the resarting mechanism and Property (iii) of Lemma B.1. It thus holds for (B.26) that

$$\frac{1}{T}\int_0^T Q^2(t)\mathrm{d}t \le \alpha^{-1}S_1 + \alpha^{1/2}\eta^{-1}S_2 + \frac{\eta^{-1}\bar{D}^2}{2T(1-\sqrt{\gamma})}. \qquad \text{(B.27)}$$

By setting $\widetilde{\phi}^{\pi_t} = \frac{1}{2}\widetilde{\phi}_0 + \frac{1}{2}\phi_0 \otimes \pi_t$ and plugging (B.27) into (4.1) of Theorem 4.1, we have

$$\frac{1}{T}\int_0^T \big(J(\pi^*) - J(\pi_t)\big)\mathrm{d}t \le \frac{\zeta}{T} + \frac{4\kappa}{T}\int_0^T Q(t)\mathrm{d}t$$
$$\le \frac{\zeta}{T} + 4\kappa\sqrt{\frac{1}{T}\int_0^T Q^2(t)\mathrm{d}t}$$
$$\le \frac{\zeta}{T} + 4\kappa\sqrt{\alpha^{-1}S_1 + \alpha^{1/2}\eta^{-1}S_2 + \frac{\eta^{-1}\bar{D}^2}{2T(1-\sqrt{\gamma})}}, \qquad \text{(B.28)}$$

where $\kappa = \big\|\widetilde{\mathcal{E}}_{\mathcal{D}_0}^{\pi^*}/\widetilde{\phi}_0\big\|_\infty$ is the concentrability coefficient and $\zeta = \mathbb{E}_{s\sim\mathcal{E}_{\pi^*}}\big[\mathrm{KL}\big(\pi^*(\cdot\,|\,s)\,\|\,\pi_0(\cdot\,|\,s)\big)\big]$ is the KL-divergence between $\pi^*$ and $\pi_0$. Here the second inequality follows from the Cauchy-Schwarz inequality. Therefore, we complete the proof of (4.9) in Theorem 4.6. In what follows, we aim to upper bound the total restarting number $N$. Recall that we have $\widetilde{W}(T_n^+) \le \bar{D}$ and $\widetilde{W}(T_{n+1}^-) \ge (\lambda-1)\bar{D}$ according to the restarting mechanism. Thus, it holds for (B.26) that

$$\frac{1}{T}\int_0^T Q^2(t)\mathrm{d}t \le \alpha^{-1}S_1 + \alpha^{1/2}\eta^{-1}S_2 - \frac{\xi}{T\eta}\big(N(\lambda-2)\bar{D}^2 - \bar{D}^2\big).$$

Since $Q^2(t) \ge 0$, it follows that

$$N \le \big((\alpha^{-1}S_1 + \alpha^{1/2}\eta^{-1}S_2)2T(1-\sqrt{\gamma})\eta + \bar{D}^2\big)\cdot\frac{\bar{D}^{-2}}{\lambda-2}$$
$$= (\lambda-2)^{-1}\big((\alpha^{-1}\eta S_1 + \alpha^{1/2}S_2)2T\bar{D}^{-2}(1-\sqrt{\gamma}) + 1\big),$$

which upper bounds the total restarting number $N$. Hence, we complete the proof of Theorem 4.6. $\qquad\square$

## C Proofs of Supporting Lemmas

In this section, we give detailed proof of supporting lemmas.

## C.1 Generality of Transition Kernel Representation

Recall that we have the following representation of the transition kernel in Assumption 4.3,

$$P(s' \,|\, s, a) = \int \widetilde{\sigma}\big((s, a, 1)^\top w\big) \varphi(s') \psi(s'; w) \mathrm{d}w, \tag{C.1}$$

where $\varphi(s') \geq 0$ and $\widetilde{\psi}(s; \cdot) \in \mathscr{P}(w)$. Such a representation has the same function class as

$$\widetilde{P}(s' \,|\, s, a) = \int \widetilde{\sigma}\big((s, a, 1)^\top w\big) \widetilde{\psi}(s'; w) \mathrm{d}w, \tag{C.2}$$

where $\widetilde{\psi}$ is a finite signed measure.

*Proof.* For simplicity, let $\mathcal{P}$ and $\widetilde{\mathcal{P}}$ denote the function class represented by (C.1) and (C.2), respectively. Note that for any transition kernel $P(s' \,|\, s, a)$ represented by (C.1), by letting $\widetilde{\psi}(s'; w) = \varphi(s')\psi(s'; w)$, such a transition kernel $P(s' \,|\, s, a)$ can be equivalently represented by $\widetilde{P}(s' \,|\, s, a)$ in (C.2). Thus, it holds that $\mathcal{P} \subseteq \widetilde{\mathcal{P}}$. Therefore, we only need to prove $\widetilde{\mathcal{P}} \subseteq \mathcal{P}$, which is equivalent to proving that for any $\widetilde{P}(s' \,|\, s, a) \in \widetilde{\mathcal{P}}$ given by (C.2), the signed measure $\widetilde{\psi}$ can be non-negative. If that is the case, by letting $\varphi(s') = \int \big(\widetilde{\psi}_+(s'; w) + \widetilde{\psi}_-(s'; -w)\big) \mathrm{d}w$ and $\psi(s'; w) = \varphi(s')^{-1} \big(\widetilde{\psi}_+(s'; w) + \widetilde{\psi}_-(s'; -w)\big)$, we can have (C.2) equivalently represented by (C.1). Note that there always exist non-negative functions $\widetilde{\psi}_+$ and $\widetilde{\psi}_-$ such that $\widetilde{\psi} = \widetilde{\psi}_+ - \widetilde{\psi}_-$. Since $\widetilde{\sigma}$ is an odd function, it holds that

$$\begin{aligned}
\widetilde{P}(s' \,|\, s, a) &= \int \widetilde{\sigma}(w^\top x) \widetilde{\psi}(s'; w) \mathrm{d}w \\
&= \int \widetilde{\sigma}(w^\top x) \widetilde{\psi}_+(s'; w) \mathrm{d}w + \int \widetilde{\sigma}(-w^\top x) \widetilde{\psi}_-(s'; w) \mathrm{d}w \\
&= \int \widetilde{\sigma}(w^\top x) \big(\widetilde{\psi}_+(s'; w) + \widetilde{\psi}_-(s'; -w)\big) \mathrm{d}w.
\end{aligned}$$

Thus, by letting $\varphi(s') = \int \big(\widetilde{\psi}_+(s'; w) + \widetilde{\psi}_-(s'; -w)\big) \mathrm{d}w$ and $\psi(s'; w) = \varphi(s')^{-1}\big(\widetilde{\psi}_+(s'; w) + \widetilde{\psi}_-(s'; -w)\big)$, it holds that $\widetilde{P}(s' \,|\, s, a) = \int \widetilde{\sigma}(w^\top x)\varphi(s')\psi(s'; \mathrm{d}w) = P(s' \,|\, s, a)$, where $\psi \in \mathscr{P}(\mathcal{W})$ and $\varphi \geq 0$. Hence, any $\widetilde{P}(s' \,|\, s, a) \in \widetilde{\mathcal{P}}$ can be equivalently represented by (C.1) and it follows that $\widetilde{\mathcal{P}} \subseteq \mathcal{P}$. Thus, (C.1) has the same function class as (C.2) and we complete the proof. $\square$

## C.2 Proof of Detailed Version of Lemma 4.4

In this section, we prove a more detailed version of Lemma 4.4, i.e., Lemma B.1.

*Proof.* We begin by a sketch of the proof of Lemma B.1. We first construct functions $Z_\pi$ and $\nu_\pi(w)$. With the use of mollifiers, we prove that there exists function $p_\pi(b)$ satisfying (C.6) and then formulate a construction of $\bar{\rho}_\pi(\theta)$, which gives way to obtain $\rho_\pi$. For the proof of Property (iii), using the technique of Talagrand's inequality and the chi-squared divergence, we establish a constant upper bound for $W_2(\rho_\pi, \rho_0)$. For the proof of Property (iv), by exploiting the inequality between $W_2$ distance and the weighted homogeneous Sobolev norm, we upper bound $W_2(\rho_{\pi_1}, \rho_{\pi_2})$ up to $O(\alpha^{-1/2})$.

**Proof of Property (i) of Lemma B.1.** We give a proof of Property (i) by a construction of $\rho_\pi$. For notational simplicity, we let $x = (s, a, 1)$. By definitions of the action value function $Q^\pi$ and the state value function $V^\pi$ in (2.2), we have

$$\begin{aligned}
Q^\pi(s, a) &= r(s, a) + \gamma \cdot \int P(s' \,|\, s, a) V^\pi(s') \mathrm{d}s' \\
&= \int \widetilde{\sigma}(w^\top x) \Big\{ B_r \cdot \mu(w) + \gamma \cdot \int \varphi(s')\psi(s'; w) V^\pi(s') \mathrm{d}s' \Big\} \mathrm{d}w \\
&= \int Z_\pi \cdot \widetilde{\sigma}(w^\top x) \nu_\pi(w) \mathrm{d}w, \tag{C.3}
\end{aligned}$$

where

$$\nu_\pi(w) = Z_\pi^{-1} \cdot \Big( B_r \mu(w) + \gamma \cdot \int \varphi(s')\psi(s';w)V^\pi(s')\mathrm{d}s' \Big), \tag{C.4}$$

$$Z_\pi = B_r + \gamma \cdot \int \varphi(s')V^\pi(s')\mathrm{d}s'. \tag{C.5}$$

Here, the second equality in (C.3) holds by (i) of Assumption 4.3. We construct $\bar\rho_\pi$ by $\bar\rho_\pi = \nu_\pi \times p_\pi$, i.e., $\bar\rho_\pi(w,b) = \nu_\pi(w)p_\pi(b)$, where $p_\pi(b)$ is defined to be a probability measure in $\mathscr{P}_2(\mathbb{R})$ such that

$$\int B_\beta \cdot \beta(b)p_\pi(\mathrm{d}b) = Z_\pi. \tag{C.6}$$

We remark that such a $p_\pi$ exists and we will provide a construction later. Since we have $V^\pi \geq 0$, $\varphi \geq 0$, and $\psi \geq 0$ by Assumption 4.3, it turns out that $\nu_\pi$ is a probability density function according to (C.4), which further suggests that $\bar\rho_\pi \in \mathscr{P}_2(\mathbb{R}^D)$. Plugging (C.6) into (C.3), it holds that

$$Q^\pi(x) = \int \sigma(x;\theta)\bar\rho_\pi(\theta)\mathrm{d}\theta,$$

where the equality holds by noting that $\sigma(x;\theta) = B_\beta \cdot \beta(b) \cdot \widetilde\sigma(w^\top x)$ in (4.2) and that $\bar\rho_\pi = \nu_\pi \times p_\pi$. Furthermore, by letting $\rho_\pi = \bar\rho_\pi + (1-\alpha^{-1})(\rho_0 - \bar\rho_\pi)$, we have

$$Q^\pi(x) = \alpha \int \sigma(x;\theta)(\rho_\pi - (1-\alpha^{-1})\rho_0)\mathrm{d}\theta = \alpha \int \sigma(x;\theta)\rho_\pi \mathrm{d}\theta = Q(x;\rho_\pi),$$

where the second equality holds by noting that $\sigma$ is odd with respect to $w$ and $b$ and that $\rho_0 \sim \mathcal{N}(0,I_D)$ is an even function. Thus, we finish the construction of $\rho_\pi$ and also complete the proof of Property (i) in Lemma B.1.

**A Construction for $p_\pi(b)$.** Recall that we have $p_\pi$ defined in (C.6). Here, we provide a construction for $p_\pi$ which has some properties that will facilitate our analysis. Ideally, we want $p_\pi$ to have global support and concentrate to its mean, which motivates us to consider $p_\pi$ to be Gaussian distribution with high variance. Recall that we assume that $\beta^{-1}$ is $\ell_\beta$-Lipschitz continuous on $[-2/3, 2/3]$ in Assumption 4.2. Let $q(b)$ be the probability density function of the standard Gaussian distribution, i.e., $q \sim \mathcal{N}(0,1)$. Then, $q_\epsilon(b-z) = \epsilon^{-1} \cdot q((b-z)/\epsilon)$ is the probability density function such that $q_\epsilon \sim \mathcal{N}(z,\epsilon^2)$. We define function $\beta_\epsilon$ as follows,

$$\beta_\epsilon(z) = \int \beta(b)q_\epsilon(b-z)\mathrm{d}b = (\beta * q_\epsilon)(z), \tag{C.7}$$

where $*$ denotes the convolution. Note that $\{q_\epsilon\}_{\epsilon>0}$ can be viewed as a class of mollifiers [25]. In particular, let

$$\bar\epsilon = \min\Big\{ \sqrt{\frac{\pi}{2}} \cdot \frac{1}{6L_{0,\beta}}, \sqrt{\frac{\pi}{2}} \cdot \frac{1}{2\ell_\beta L_{1,\beta}}, 1 \Big\}, \tag{C.8}$$

where $L_{0,\beta}$ and $L_{1,\beta}$ characterize the Lipschitz continuity and smoothness of $\beta$ respectively and $\beta^{-1}$ is $\ell_\beta$-Lipschitz continuous in $[-2/3, 2/3]$ by Assumption 4.2. For the approximation error of mollifier $\beta_\epsilon$, it holds that

$$\begin{aligned}
\big|\beta(z) - \beta_{\bar\epsilon}(z)\big| &= \Big| \int (\beta(z) - \beta(z'))q_{\bar\epsilon}(z-z')\mathrm{d}z' \Big| \\
&\leq L_{0,\beta} \cdot \int |z-z'|q_{\bar\epsilon}(z-z')\mathrm{d}z' \\
&= L_{0,\beta} \cdot \bar\epsilon \cdot \sqrt{\frac{2}{\pi}},
\end{aligned}$$

where the last inequality follows from $\int |z|q_\epsilon(z)\mathrm{d}z = \epsilon \cdot \sqrt{2/\pi}$. Similarly, we have $|\dot\beta(z) - \dot\beta_{\bar\epsilon}(z)| \leq L_{1,\beta} \cdot \bar\epsilon \cdot \sqrt{2/\pi}$. By definition of $\bar\epsilon$ in (C.8), it further holds that

$$\sup_{b\in\beta^{-1}([-2/3,2/3])} \big|\beta(b) - \beta_{\bar\epsilon}(b)\big| \leq 1/6, \tag{C.9}$$

$$\sup_{b\in\beta^{-1}([-2/3,2/3])} \big|\dot\beta(b) - \dot\beta_{\bar\epsilon}(b)\big| \leq \frac{1}{2\ell_\beta}. \tag{C.10}$$

Note that $\beta(b)$ is a monotonic function with $|\dot{\beta}(b)| \geq 1/\ell_\beta$ in $\beta^{-1}([-2/3, 2/3])$ by Assumption 4.2. With regard to (C.10), it follows that $\beta_{\bar{\epsilon}}(b)$ is also monotonic in $\beta^{-1}([-2/3, 2/3])$ and that

$$|\dot{\beta}_{\bar{\epsilon}}(b)| \geq |\dot{\beta}(b)| - |\dot{\beta}(b) - \dot{\beta}_{\bar{\epsilon}}(b)| \geq \frac{1}{2\ell_\beta}, \quad \forall b \in \beta^{-1}([-2/3, 2/3]). \tag{C.11}$$

Furthermore, by (C.9) and the continuity of $\beta_{\bar{\epsilon}}$ in $\beta^{-1}([-2/3, 2/3])$, we have

$$[-1/2, 1/2] \subseteq \beta_{\bar{\epsilon}}(\beta^{-1}([-2/3, 2/3])). \tag{C.12}$$

The monotonicity of $\beta_{\bar{\epsilon}}(b)$, (C.11), and (C.12) together show that $\beta_{\bar{\epsilon}}^{-1}$ exists and is $2\ell_\beta$-Lipschitz continuous in $[-1/2, 1/2]$. Moreover, since $\beta$ is an odd function, it holds by (C.7) that $\beta_{\bar{\epsilon}}$ is also an odd function with $\beta_{\bar{\epsilon}}(0) = 0$. Hence, it holds that $\beta_{\bar{\epsilon}}^{-1}([-1/2, 1/2]) \subseteq [-\ell_\beta, \ell_\beta]$. Furthermore, by (C.5), it holds that

$$Z_\pi = B_r + \gamma \cdot \int \varphi(s') V^\pi(s') \mathrm{d}s'$$

$$\leq B_r + \gamma \cdot (1-\gamma)^{-1} \cdot B_r \cdot \int \varphi(s') \mathrm{d}s'$$

$$\leq B_r + \gamma(1-\gamma)^{-1} B_r M_{1,\varphi},$$

where the first inequality follows from the fact that $V^\pi(s) \leq (1-\gamma)^{-1} \cdot \sup_{s,a} r(s,a)$ and the last inequality follows from Assumption 4.3 that $\int \varphi(s') \mathrm{d}s' \leq M_{1,\varphi}$. By setting $B_\beta \geq 2(B_r + \gamma(1-\gamma)^{-1} B_r M_{1,\varphi})$, it holds that $|Z_\pi/B_\beta| \leq 1/2$, which indicates that $\beta_{\bar{\epsilon}}^{-1}(Z_\pi/B_\beta)$ exists and allows $p_\pi(b) = q_{\bar{\epsilon}}(b - \beta_{\bar{\epsilon}}^{-1}(Z_\pi/B_\beta))$ to be the probability density function such that $p_\pi(b) \sim \mathcal{N}(\beta_{\bar{\epsilon}}^{-1}(Z_\pi/B_\beta), \bar{\epsilon}^2)$. For the mean value $\beta_{\bar{\epsilon}}^{-1}(Z_\pi/B_\beta)$, recalling that $\beta_{\bar{\epsilon}}^{-1}([-1/2, 1/2]) \subseteq [-\ell_\beta, \ell_\beta]$, it thus holds that

$$\left|\beta_{\bar{\epsilon}}^{-1}(Z_\pi/B_\beta)\right| \leq \ell_\beta. \tag{C.13}$$

Following from (C.7), we have

$$\int \beta(b) p_\pi(b) \mathrm{d}b = \int \beta(b) q_{\bar{\epsilon}}(b - \beta_{\bar{\epsilon}}^{-1}(Z_\pi/B_\beta)) \mathrm{d}b = \beta_{\bar{\epsilon}}(\beta_{\bar{\epsilon}}^{-1}(Z_\pi/B_\beta)) = Z_\pi/B_\beta.$$

Hence, our construction of $p_\pi$ here is in line with the definition of $p_\pi$ in (C.6). In the sequel, we consider $p_\pi(b) = q_{\bar{\epsilon}}(b - \beta_{\bar{\epsilon}}^{-1}(Z_\pi/B_\beta))$ to hold all along.

**Proof of Property (ii) of Lemma B.1.** Here we show that $g(\cdot; \pi, \rho_\pi) = 0$ is a direct result of $Q^\pi(x) = Q(x; \rho_\pi)$ in Property (i). Note that

$$Q^\pi(x) - r(x) - \gamma \mathbb{E}_{x' \sim \widetilde{P}^\pi(\cdot \mid x)} Q^\pi(x') = 0 \tag{C.14}$$

holds by the definition of the action value function $Q^\pi$ in (2.2) for any $x \in \mathcal{S} \times \mathcal{A}$. Since we have $Q^\pi(x) = Q(x; \rho_\pi)$ proved, by plugging (C.14) into the definition of $g$ in (3.6), where $Q(\cdot; \rho_\pi)$ is substituted for $Q^\pi$, it follows that $g(x; \pi, \rho_\pi) = 0$. Thus, we complete the proof of Property (ii) of Lemma B.1.

**Proof of Property (iii) of Lemma B.1.**

In what follows, we aim to upper bound $W^2(\rho_\pi, \rho_0)$. We summerize our aforementioned constructions as follows,

$$Z_\pi = B_r + \gamma \cdot \int \varphi(s') V^\pi(s') \mathrm{d}s',$$

$$\nu_\pi(w) = Z_\pi^{-1}\left(B_r \mu(w) + \gamma \cdot \int \varphi(s') \psi(s'; w) V^\pi(s') \mathrm{d}s'\right), \tag{C.15}$$

$$p_\pi(b) = q_{\bar{\epsilon}}(b - \beta_{\bar{\epsilon}}^{-1}(Z_\pi/B_\beta)), \tag{C.16}$$

$$\bar{\rho}_\pi(\theta) = p_\pi(b) \nu_\pi(w), \tag{C.17}$$

$$\rho_\pi = \bar{\rho}_\pi + (1 - \alpha^{-1})(\rho_0 - \bar{\rho}_\pi). \tag{C.18}$$

Plugging (C.16) into the definition of Chi-squared divergence, we have

$$\chi^2\big(p_\pi \,\|\, \rho_{0,b}\big) = \int \frac{p_\pi^2}{\rho_{0,b}} \mathrm{d}b - 1 = \frac{1}{\bar{\epsilon}\sqrt{2 - \bar{\epsilon}^2}} \cdot \exp\left\{ \frac{\big(\beta_{\bar{\epsilon}}^{-1}(Z_\pi/B_\beta)\big)^2}{2 - \bar{\epsilon}^2} \right\} - 1, \qquad \text{(C.19)}$$

Note that we have $\bar{\epsilon} = \min\left\{ \sqrt{\pi/2} \cdot (6L_{0,\beta})^{-1}, \sqrt{\pi/2} \cdot (2\ell_\beta L_{1,\beta})^{-1}, 1 \right\}$ by (C.8) and $\big|\beta_{\bar{\epsilon}}^{-1}(Z_\pi/B_\beta)\big| \le \ell_\beta$ by (C.13). Hence, we have $\chi^2(p_\pi \,\|\, \rho_{0,b})$ upper bounded. As for $\nu_\pi$ in (C.15), we have

$$\chi^2(\nu_\pi \,\|\, \rho_{0,w}) = \chi^2\bigg( B_r Z_\pi^{-1}\mu + \int V^\pi(s') Z_\pi^{-1}\varphi(s')\psi(s';w)\mathrm{d}s' \,\Big\|\, \rho_{0,w} \bigg)$$
$$\le 3\bigg( \chi^2(B_r Z_\pi^{-1}\mu \,\|\, \rho_{0,w}) + \chi^2\bigg( \int V^\pi(s') Z_\pi^{-1}\varphi(s')\psi(s';\cdot)\mathrm{d}s' \,\Big\|\, \rho_{0,w} \bigg) + 1 \bigg),$$
$$\text{(C.20)}$$

where the inequality holds by Property (iii) of Lemma D.3. For the first term on the right-hand side of (C.20), we have

$$\chi^2(B_r Z_\pi^{-1}\mu \,\|\, \rho_{0,w}) = B_r^2 Z_\pi^{-2}\chi^2(\mu \,\|\, \rho_{0,w}) + (1 - B_r Z_\pi^{-1})^2$$
$$\le \chi^2(\mu \,\|\, \rho_{0,w}) + 1$$
$$\le M_\mu + 1, \qquad \text{(C.21)}$$

where the equality holds by Property (i) of Lemma D.3, the first inequality holds by noting that $|B_r Z_\pi^{-1}| \le 1$ and the last inequality follows from $\chi^2(\mu \,\|\, \rho_{0,w}) < M_\mu$ by Assumption 4.3. Hence, the first term on the right-hand side of (C.20) is upper bounded. As for the second term, by Property (iv) of Lemma D.3, we have

$$\chi^2\bigg( \int V^\pi(s') Z_\pi^{-1}\varphi(s')\psi(s';\cdot)\mathrm{d}s' \,\Big\|\, \rho_{0,w} \bigg)$$
$$\le \int \big(V^\pi(s') Z_\pi^{-1}\varphi(s')\big)^2\mathrm{d}s' \cdot \int \chi^2(\psi(s';\cdot) \,\|\, \rho_{0,w})\mathrm{d}s' + \bigg( \int V^\pi(s') Z_\pi^{-1}\varphi(s')\mathrm{d}s' - 1 \bigg)^2$$
$$\le (1-\gamma)^{-2}M_{2,\varphi} \cdot M_\psi + (1-\gamma)^{-2}M_{1,\varphi}^2 + 1, \qquad \text{(C.22)}$$

where the last inequality holds by noting that $|V^\pi| \le (1-\gamma)^{-1}B_r$, $Z_\pi \ge B_r$, $\|\mathcal{S}\| \le 1$, and that $\chi^2\big(\psi(s';\cdot) \,\|\, \rho_{0,w}\big) < M_\psi$ by Assumption 4.3. Hence, it holds that the second term of (C.20) is also upper bounded. Plugging (C.21) and (C.22) into (C.20), we can establish the upper bound for $\chi^2(\nu_\pi \,\|\, \rho_{0,w})$. Furthermore, by Property (ii) of Lemma D.3 and noting that $\bar{\rho}_\pi = p_\pi \times \nu_\pi$, we have $\chi^2(\bar{\rho}_\pi \,\|\, \rho_0)$ upper bounded as well, that is,

$$\chi^2(\bar{\rho}_\pi \,\|\, \rho_0) < \frac{1}{2}\bar{D}^2,$$

where $\bar{D}$ depends on absolute constants occurring in (C.19), (C.21), and (C.22), i.e., $\ell_\beta, L_{0,\beta}, L_{1,\beta}, B_r, M_{1,\varphi}, M_{2,\varphi}, M_\mu, M_\psi$ in Assumptions 4.2 and 4.3. Since $\rho_0 \sim \mathcal{N}(0, I_D)$, it holds for any $\rho_\pi$ that,

$$\frac{1}{2}W_2(\rho_\pi, \rho_0)^2 \le \mathrm{KL}(\rho_\pi \,\|\, \rho_0) \le \int \Big( \frac{\rho_\pi}{\rho_0} - 1 \Big)\frac{\rho_\pi}{\rho_0}\rho_0(\mathrm{d}\theta) = \int \Big( \frac{\rho_\pi}{\rho_0} - 1 \Big)^2 \rho_0(\mathrm{d}\theta)$$
$$= \int \Big( \frac{(1-\alpha^{-1})\rho_0(\theta) + \alpha^{-1}\bar{\rho}_\pi(\theta)}{\rho_0(\theta)} - 1 \Big)^2 \rho_0(\mathrm{d}\theta) = \alpha^{-2}\chi^2(\bar{\rho}_\pi \,\|\, \rho_0) < \frac{\alpha^{-2}}{2}\bar{D}^2,$$
$$\text{(C.23)}$$

where the first inequality follows from Talagrand's inequality in Lemma E.4. Plugging $\widetilde{W}_2 = \alpha W_2$ into (C.23), we complete the proof of Property (iii) of Lemma B.1.

**Proof of Property (iv) of Lemma B.1.**

**Upper Bounding $W_2(p_{\pi_1}, p_{\pi_2})$.** Following the property of $W_2$ distance with respect to Gaussian distribution, we have $W_2(p_{\pi_1}, p_{\pi_2}) = \big|\beta_{\bar{\epsilon}}^{-1}(Z_{\pi_1}/B_\beta) - \beta_{\bar{\epsilon}}^{-1}(Z_{\pi_2}/B_\beta)\big|$. Recall that we have $|Z_\pi/B_\beta| \leq 1/2$ and that $\beta_{\bar{\epsilon}}^{-1}$ is $2\ell_\beta$-Lipschitz continuous on $[-1/2, 1/2]$. It then holds that

$$W_2(p_{\pi_1}, p_{\pi_2}) = \big|\beta_{\bar{\epsilon}}^{-1}(Z_{\pi_1}/B_\beta) - \beta_{\bar{\epsilon}}^{-1}(Z_{\pi_2}/B_\beta)\big| \leq 2\ell_\beta \cdot |Z_{\pi_1} - Z_{\pi_2}|/B_\beta. \tag{C.24}$$

Meanwhile, we have

$$\begin{aligned}
|Z_{\pi_1} - Z_{\pi_2}| &\leq \gamma \cdot \int \varphi(s') \cdot \big|V^{\pi_1}(s') - V^{\pi_2}(s')\big| \mathrm{d}s' \\
&\leq \gamma \cdot \int \varphi(s') \mathrm{d}s' \cdot \sup_{s' \in \mathcal{S}} \big|V^{\pi_1}(s') - V^{\pi_2}(s')\big| \\
&\leq \gamma \cdot M_{1,\varphi} \cdot \sup_{s' \in \mathcal{S}} \big|V^{\pi_1}(s') - V^{\pi_2}(s')\big|,
\end{aligned} \tag{C.25}$$

where the last inequality holds by (ii) of Assumption 4.3. Plugging (C.25) into (C.24), it holds for $W_2(p_{\pi_1}, p_{\pi_2})$ that

$$W_2(p_{\pi_1}, p_{\pi_2}) \leq \frac{2\ell_\beta \cdot \gamma \cdot M_{1,\varphi} \cdot \sup_{s' \in \mathcal{S}} \big|V^{\pi_1}(s') - V^{\pi_2}(s')\big|}{B_\beta}. \tag{C.26}$$

**Upper Bounding $W_2(\nu_{\pi_1}, \nu_{\pi_2})$.** By definition of $\nu_\pi$ in (C.15), we have

$$\nu_{\pi_1} - \nu_{\pi_2} = \frac{B_r(Z_{\pi_2} - Z_{\pi_1})}{Z_{\pi_1} Z_{\pi_2}} \mu + \gamma \int \varphi(s') \psi(s'; \cdot) \Big(\frac{V^{\pi_1}}{Z_{\pi_1}} - \frac{V^{\pi_2}}{Z_{\pi_2}}\Big) \mathrm{d}s'. \tag{C.27}$$

For $V^{\pi_1}/Z_{\pi_1} - V^{\pi_2}/Z_{\pi_2}$, it holds that

$$\begin{aligned}
\Big|\frac{V^{\pi_1}}{Z_{\pi_1}} - \frac{V^{\pi_2}}{Z_{\pi_2}}\Big| &\leq \max\{V^{\pi_1}, V^{\pi_2}\} \cdot \frac{|Z_{\pi_1} - Z_{\pi_2}|}{Z_{\pi_1} Z_{\pi_2}} + \max\{Z_{\pi_1}^{-1}, Z_{\pi_2}^{-1}\} |V^{\pi_1} - V^{\pi_2}| \\
&\leq (B_r^{-1}(1-\gamma)^{-1} \gamma M_{1,\varphi} + B_r^{-1}) \sup_{s' \in S} \big|V^{\pi_1}(s') - V^{\pi_2}(s')\big|,
\end{aligned} \tag{C.28}$$

where the last inequality holds by noting that

$$\Big|\frac{Z_{\pi_2} - Z_{\pi_1}}{Z_{\pi_1} Z_{\pi_2}}\Big| \leq \gamma M_{1,\varphi} B_r^{-2} \sup_{s' \in S} \big|V^{\pi_1}(s') - V^{\pi_2}(s')\big|. \tag{C.29}$$

Here the inequality in (C.29) holds by (C.25) and the fact that $Z_\pi \geq B_r$. For $W_2(\nu_{\pi_1}, \nu_{\pi_2})$, by Lemma E.5, it holds that

$$W_2(\nu_{\pi_1}, \nu_{\pi_2}) \leq 2\|\nu_{\pi_1} - \nu_{\pi_2}\|_{\dot{H}^{-1}(\nu_{\pi_1})}. \tag{C.30}$$

Recall that we have $\nu_\pi$ defined in (C.15) that

$$\nu_\pi(w) = Z_\pi^{-1} \cdot \Big(B_r \mu(w) + \gamma \cdot \int \varphi(s') \psi(s'; w) V^\pi(s') \mathrm{d}s'\Big),$$

$$Z_\pi = B_r + \gamma \cdot \int \varphi(s') V^\pi(s') \mathrm{d}s'$$

where $Z_\pi^{-1}(B_r + \gamma \cdot \int \varphi(s') V^\pi(s') \mathrm{d}s') = 1$, $B_r Z_\pi^{-1} \geq 0$, and $\gamma \varphi(s') V^\pi(s') Z_\pi^{-1} \geq 0$, which indicate that $\nu_\pi$ is in the convex hull of $\mu$ and $\psi(s'; \cdot)$. Hence, by Property (i) of Lemma D.4, it holds that

$$2\|\nu_{\pi_1} - \nu_{\pi_2}\|_{\dot{H}^{-1}(\nu_{\pi_1})} \leq 2 \max\Big\{\sup_s \|\nu_{\pi_1} - \nu_{\pi_2}\|_{\dot{H}^{-1}(\psi(s; \cdot))}, \|\nu_{\pi_1} - \nu_{\pi_2}\|_{\dot{H}^{-1}(\mu)}\Big\}. \tag{C.31}$$

Furthermore, following from (C.27) and $Z_\pi^{-1}(B_r + \gamma \cdot \int \varphi(s') V^\pi(s') \mathrm{d}s') = 1$, by Property (ii) of Lemma D.4, it holds for $2\|\nu_{\pi_1} - \nu_{\pi_2}\|_{\dot{H}^{-1}(\mu)}$ that

$$\begin{aligned}
2\|\nu_{\pi_1} - \nu_{\pi_2}\|_{\dot{H}^{-1}(\mu)} &\leq \max\Big\{\sup_{s' \in \mathcal{S}} \|\mu - \psi(s'; \cdot)\|_{\dot{H}^{-1}(\mu)}, \sup_{(s', s'') \in \mathcal{S} \times \mathcal{S}} \|\psi(s'; \cdot) - \psi(s''; \cdot)\|_{\dot{H}^{-1}(\mu)}\Big\} \\
&\quad \cdot \Big(\Big|\frac{B_r(Z_{\pi_2} - Z_{\pi_1})}{Z_{\pi_1} Z_{\pi_2}}\Big| + \gamma \int \varphi(s') \Big|\frac{V^{\pi_1}}{Z_{\pi_1}} - \frac{V^{\pi_2}}{Z_{\pi_2}}\Big| \mathrm{d}s'\Big).
\end{aligned} \tag{C.32}$$

Plugging (iii) of Assumption 4.3, (C.28), and (C.29) into (C.32), we have

$$2\|\nu_{\pi_1} - \nu_{\pi_2}\|_{\dot{H}^{-1}(\mu)} \leq \mathcal{G} \cdot \left( \gamma M_{1,\varphi} B_r^{-1} + \gamma M_{1,\varphi} \left( B_r^{-1}(1-\gamma)^{-1} \gamma M_{1,\varphi} + B_r^{-1} \right) \right)$$
$$\cdot \sup_{s' \in S} \left| V^{\pi_1}(s') - V^{\pi_2}(s') \right|. \tag{C.33}$$

By simply substituting $\|\cdot\|_{\dot{H}^{-1}(\psi(s;))}$ for $\|\cdot\|_{\dot{H}^{-1}(\nu)}$ in both (C.32) and (C.33), it also holds for $2\|\nu_{\pi_1} - \nu_{\pi_2}\|_{\dot{H}^{-1}(\psi(s;\cdot))}$ that

$$2\|\nu_{\pi_1} - \nu_{\pi_2}\|_{\dot{H}^{-1}(\psi(s;\cdot))} \leq \mathcal{G} \cdot \left( \gamma M_{1,\varphi} B_r^{-1} + \gamma M_{1,\varphi} \left( B_r^{-1}(1-\gamma)^{-1} \gamma M_{1,\varphi} + B_r^{-1} \right) \right)$$
$$\cdot \sup_{s' \in S} \left| V^{\pi_1}(s') - V^{\pi_2}(s') \right|. \tag{C.34}$$

Combining (C.30), (C.31), (C.33), and (C.34), we have

$$W_2(\nu_{\pi_1}, \nu_{\pi_2}) \leq \mathcal{G} \cdot \left( \gamma M_{1,\varphi} B_r^{-1} + \gamma M_{1,\varphi} \left( B_r^{-1}(1-\gamma)^{-1} \gamma M_{1,\varphi} + B_r^{-1} \right) \right)$$
$$\cdot \sup_{s' \in S} \left| V^{\pi_1}(s') - V^{\pi_2}(s') \right|. \tag{C.35}$$

**Upper Bounding** $W_2(\rho_{\pi_1}, \rho_{\pi_2})$. Note that we have $\bar{\rho}_\pi = \nu_\pi \times p_\pi$ in (C.17). By Lemma D.5, we have

$$W_2(\bar{\rho}_{\pi_1}, \bar{\rho}_{\pi_2}) \leq \sqrt{W_2^2(\nu_{\pi_1}, \nu_{\pi_2}) + W_2^2(p_{\pi_1}, p_{\pi_2})} \leq \mathcal{B}' \sup_{s \in S} \left| V^{\pi_1}(s') - V^{\pi_2}(s') \right|, \tag{C.36}$$

where $\mathcal{B}'$ depends on the discount factor $\gamma$ and absolute constants $\ell_\beta, B_\beta, B_r, M_{1,\varphi}, \mathcal{G}$ in Assumption 4.2 and 4.3 according to (C.26) and (C.35). By performance difference lemma [33], we have

$$\left| V^{\pi_1}(s) - V^{\pi_2}(s) \right| = (1-\gamma)^{-1} \cdot \left| \mathbb{E}_{s' \sim \mathcal{E}_s^{\pi_1}} \left[ \left\langle A^{\pi_2}(s', \cdot), \pi_1(\cdot \,|\, s') - \pi_2(\cdot \,|\, s') \right\rangle_{\mathcal{A}} \right] \right|$$
$$= (1-\gamma)^{-1} \cdot \left| \mathbb{E}_{s' \sim \mathcal{E}_s^{\pi_1}} \left[ \left\langle Q^{\pi_2}(s', \cdot), \pi_1(\cdot \,|\, s') - \pi_2(\cdot \,|\, s') \right\rangle_{\mathcal{A}} \right] \right|$$
$$\leq (1-\gamma)^{-2} \cdot B_r \cdot \mathbb{E}_{s' \sim \mathcal{E}_s^{\pi_1}} \left[ \left\| \pi_1(\cdot \,|\, s') - \pi_2(\cdot \,|\, s') \right\|_1 \right]. \tag{C.37}$$

Here the inequality follows from $|Q^\pi(s, a)| \leq (1-\gamma)^{-1} \cdot B_r$. We let $\mathcal{B} = \mathcal{B}' B_r (1-\gamma)^{-2}$. Plugging (C.37) into (C.36), we have

$$W_2(\bar{\rho}_{\pi_1}, \bar{\rho}_{\pi_2}) \leq \mathcal{B} \cdot \sup_{s \in \mathcal{S}} \mathbb{E}_{s' \sim \mathcal{E}_s^{\pi_1}} \left[ \left\| \pi_1(\cdot \,|\, s') - \pi_2(\cdot \,|\, s') \right\|_1 \right].$$

Recall that we have $\rho_\pi = \bar{\rho}_\pi + (1-\alpha^{-1})(\rho_0 - \bar{\rho}_\pi)$ in (C.18). Then, by Lemma D.6, it holds that

$$W_2(\rho_{\pi_1}, \rho_{\pi_2}) \leq \alpha^{-\frac{1}{2}} W_2(\bar{\rho}_{\pi_1}, \bar{\rho}_{\pi_2}) \leq \alpha^{-\frac{1}{2}} \mathcal{B} \cdot \sup_{s \in \mathcal{S}} \mathbb{E}_{s' \sim \mathcal{E}_s^{\pi_1}} \left[ \left\| \pi_1(\cdot \,|\, s') - \pi_2(\cdot \,|\, s') \right\|_1 \right],$$

which completes the proof of Lemma B.1. □

# D Technical Results

In this section, we state and prove some technical results used in the proof of main theorems and lemmas.

**Lemma D.1.** Under Assumptions 4.3 and 4.2, it holds for any $\rho \in \mathscr{P}_2(\mathbb{R}^D)$ that

$$\sup_{x \in \mathcal{X}} \left| Q(x; \rho) \right| \leq \alpha \cdot B_1 \cdot W_2(\rho, \rho_0), \tag{D.1}$$

$$\sup_{\theta \in \mathbb{R}^D} \left\| \nabla_\theta g(\theta; \rho) \right\|_{\mathrm{F}} \leq \alpha^{-1} \cdot B_2 \cdot \left( 2\alpha \cdot B_1 \cdot W_2(\rho, \rho_0) + B_r \right). \tag{D.2}$$

*Proof.* Following from Assumptions 4.3 and 4.2, we have that $\|\nabla_\theta \sigma(x; \theta)\| \le B_1$ for any $x \in \mathcal{X}$ and $\theta \in \mathbb{R}^D$, which implies that $\mathrm{Lip}(\sigma(x; \cdot)/B_1) \le 1$ for any $x \in \mathcal{X}$. Note that $Q(x; \rho_0) = 0$ for any $x \in \mathcal{X}$. Thus, by (A.7) and the inequality between $W_1$ distance and $W_2$ distance [58], we have for any $\rho \in \mathscr{P}_2(\mathbb{R}^D)$ and $x \in \mathcal{X}$ that

$$\big|Q(x; \rho)\big| = \alpha \cdot \left| \int \sigma(x; \theta) \cdot \mathrm{d}(\rho - \rho_0)(\theta) \right| \le \alpha \cdot B_1 \cdot W_1(\rho, \rho_0) \le \alpha \cdot B_1 \cdot W_2(\rho, \rho_0). \quad \text{(D.3)}$$

which completes the proof of (D.1) in Lemma D.1. Following from the definition of $g$ in (3.6), we have for any $x \in \mathcal{X}$ and $\rho \in \mathscr{P}_2(\mathbb{R}^D)$ that

$$\big\|\nabla_\theta g(\theta; \rho)\big\|_{\mathrm{F}} \le \alpha^{-1} \cdot \mathbb{E}_{\widetilde{\mathcal{D}}}\Big[\big|Q(x; \rho) - r - \gamma \cdot Q(x'; \rho)\big| \cdot \big\|\nabla_{\theta\theta}^2 \sigma(x; \theta)\big\|_{\mathrm{F}}\Big]$$
$$\le \alpha^{-1} \cdot B_2 \cdot \big(2\alpha \cdot B_1 \cdot W_2(\rho, \rho_0) + B_r\big).$$

Here the last inequality follows from (D.3) and the fact that $\|\nabla_{\theta\theta}^2 \sigma(x; \theta)\|_{\mathrm{F}} \le B_2$ for any $x \in \mathcal{X}$ and $\rho \in \mathscr{P}_2(\mathbb{R}^D)$, which follows from Assumptions 4.3 and 4.2. Thus, we complete the proof of Lemma D.1. $\qquad\square$

**Lemma D.2.** For $\rho_0, \rho_t, \rho_{\pi_t} \in \mathscr{P}_2(\mathbb{R}^D)$ and the geodesic $\alpha_t^{[0,1]}$ connecting $\rho_t$ and $\rho_{\pi_t}$, we have

$$\sup_{s \in [0,1]} W_2(\alpha_t^s, \rho_0) \le 2\max\{W_2(\rho_{\pi_t}, \rho_0), W_2(\rho_t, \rho_0)\}. \quad \text{(D.4)}$$

*Proof.* We give a proof by contradiction. Note that $\alpha_t^s$ is the geodesic connecting $\rho_{\pi_t}$ and $\rho_t$. Assume there exists $t$ such that

$$\sup_{s \in [0,1]} W_2(\alpha_t^s, \rho_0) > 2\max\{W_2(\rho_{\pi_t}, \rho_0), W_2(\rho_t, \rho_0)\}.$$

Then, according to the triangle inequality of $W_2$ metric [59], we have

$$W_2(\rho_t, \alpha_t^s) \ge |W_2(\alpha_t^s, \rho_0) - W_2(\rho_t, \rho_0)| > |2W_2(\rho_t, \rho_0) - W_2(\rho_t, \rho_0)| = W_2(\rho_t, \rho_0),$$

and $W_2(\rho_{\pi_t}, \alpha_t^s) > W_2(\rho_{\pi_t}, \rho_0)$ for the same sake, which conflicts with the definition of geodesic that $W_2(\rho_t, \alpha_t^s) + W_2(\alpha_t^s, \rho_{\pi_t}) = W_2(\rho_t, \rho_{\pi_t}) \le W_2(\rho_t, \rho_0) + W_2(\rho_{\pi_t}, \rho_0)$. Hence, such $t$ does not exist and (D.4) holds. Thus we complete the proof of Lemma D.2. $\qquad\square$

**Lemma D.3.** The Chi-squared divergence has the following properties.

(i) For any probability measure $g \in \mathscr{P}(\Theta)$, function $f : \Theta \to \mathbb{R}$, and $\alpha \in \mathbb{R}$, we have

$$\chi^2(\alpha f \,\|\, g) = \alpha^2 \chi^2(f \,\|\, g) + (1 - \alpha)^2.$$

(ii) For any probability measures $g_1 \in \mathscr{P}(\Theta_1)$ and $g_2 \in \mathscr{P}(\Theta_2)$, functions $f_1 : \Theta_1 \to \mathbb{R}$ and $f_2 : \Theta_2 \to \mathbb{R}$, we have

$$\chi^2(f_1 \times f_2 \,\|\, g_1 \times g_2) = \chi^2(f_1 \,\|\, g_1) \cdot \chi^2(f_2 \,\|\, g_2) + \chi^2(f_1 \,\|\, g_1) + \chi^2(f_2 \,\|\, g_2),$$

where $f_1 \times f_2$ is the product of $f_1$ and $f_2$, and $g_1 \times g_2$ is the product measure of $g_1$ and $g_2$, i.e., $(f_1 \times f_2)(\theta_1 \times \theta_2) = f_1(\theta_1) f_2(\theta_2)$ and $(g_1 \times g_2)(\theta_1 \times \theta_2) = g_1(\theta_1) g_2(\theta_2)$, respectively.

(iii) For any probability measure $g \in \mathscr{P}(\Theta)$, functions $f_1 : \Theta \to \mathbb{R}$ and $f_2 : \Theta \to \mathbb{R}$, we have

$$\chi^2(f_1 + f_2 \,\|\, g) \le 3\big(\chi^2(f_1 \,\|\, g) + \chi^2(f_2 \,\|\, g) + 1\big).$$

(iv) For any probability measure $g \in \mathscr{P}(\Theta)$, function $f : \mathcal{X} \times \Theta \to \mathbb{R}$ and $\alpha : \mathcal{X} \to \mathbb{R}$, we have

$$\chi^2\left(\int \alpha(x) f(d, \cdot) dx \,\Big\|\, g\right) \le \int \alpha(x)^2 \mathrm{d}x \cdot \int \chi^2(f(x, \cdot) \,\|\, g) \mathrm{d}x + \left(\int \alpha(x) \mathrm{d}x - 1\right)^2.$$

*Proof.* **Proof of Property (i) of Lemma D.3.** By definition of the Chi-squared divergence, we have

$$\chi^2(\alpha f \| g) = \int \left(\frac{\alpha f}{g} - 1\right)^2 \mathrm{d}g$$

$$= \int \left(\alpha\left(\frac{f}{g} - 1\right) + \alpha - 1\right)^2 \mathrm{d}g$$

$$= \alpha^2 \chi^2(f \| g) + (\alpha - 1)^2.$$

Thus, we complete the proof of Property (i) of Lemma D.3.

**Proof of Property (ii) of Lemma D.3.** Let $\widetilde{f}_1 = f_1/g_1 - 1$ and $\widetilde{f}_2 = f_2/g_2 - 1$. It then holds that

$$\chi^2(f_1 \times f_2 \| g_1 \times g_2) = \int \left(\frac{f_1 \times f_2}{g_1 \times g_2} - 1\right)^2 \mathrm{d}(g_1 \times g_2) = \int (\widetilde{f}_1 \times \widetilde{f}_2 + \widetilde{f}_1 + \widetilde{f}_2)^2 \mathrm{d}(g_1 \times g_2).$$

By further noting that $\int \widetilde{f}_1 \mathrm{d}g_1 = 0$, $\int \widetilde{f}_2 \mathrm{d}g_2 = 0$, $\int \widetilde{f}_1^2 \mathrm{d}g_1 = \chi^2(f_1 \| g_1)$, and $\int \widetilde{f}_2^2 \mathrm{d}g_2 = \chi^2(f_2 \| g_2)$, we have

$$\chi^2(f_1 \times f_2 \| g_1 \times g_2)$$

$$= \int \left(\widetilde{f}_1^2 \times \widetilde{f}_2^2 + \widetilde{f}_1^2 + \widetilde{f}_2^2 + 2(\widetilde{f}_1)^2 \times \widetilde{f}_2 + 2(\widetilde{f}_2)^2 \times \widetilde{f}_1 + 2\widetilde{f}_1 \times \widetilde{f}_2\right) \mathrm{d}(g_1 \times g_2)$$

$$= \chi^2(f_1 \| g_1) \cdot \chi^2(f_2 \| g_2) + \chi^2(f_1 \| g_1) + \chi^2(f_2 \| g_2).$$

Thus, we complete the proof of Property (ii) of Lemma D.3.

**Proof of Property (iii) of Lemma D.3.** Let $\widetilde{f}_1 = f_1/g_1 - 1$ and $\widetilde{f}_2 = f_2/g_2 - 1$. It then holds that

$$\chi^2(f_1 + f_2 \| g) = \int \left(\frac{f_1 + f_2}{g} - 1\right)^2 \mathrm{d}g = \int (\widetilde{f}_1 + \widetilde{f}_2 + 1)^2 \mathrm{d}g.$$

By further noting that $\int \widetilde{f}_1^2 \mathrm{d}g_1 = \chi^2(f_1 \| g_1)$ and $\int \widetilde{f}_2^2 \mathrm{d}g_2 = \chi^2(f_2 \| g_2)$, we have

$$\chi^2(f_1 + f_2 \| g) \le 3 \int \left((\widetilde{f}_1)^2 + (\widetilde{f}_2)^2 + 1\right) \mathrm{d}g$$

$$= 3\left(\chi^2(f_1 \| g) + \chi^2(f_2 \| g) + 1\right),$$

where the inequality follows from the Cauchy-Schwarz inequality. Thus, we complete the proof of Property (iii) of Lemma D.3.

**Proof of Property (iv) of Lemma D.3.** Let $\widetilde{f}(x, \cdot) = f(x, \cdot)/g(\cdot) - 1$. It then holds that

$$\chi^2\left(\int \alpha(x) f(x, \cdot) \mathrm{d}x \| g\right) = \int \left(\int \alpha(x) \widetilde{f}(x, \theta) \mathrm{d}x\right)^2 g(\mathrm{d}\theta) + \left(\int \alpha(x) \mathrm{d}x - 1\right)^2$$

$$\le \int \left(\int \alpha(x)^2 \mathrm{d}x \cdot \int \widetilde{f}(x, \theta)^2 \mathrm{d}x\right) g(\mathrm{d}\theta) + \left(\int \alpha(x) \mathrm{d}x - 1\right)^2$$

$$= \int \alpha(x)^2 \mathrm{d}x \cdot \int \chi^2(f(x, \cdot) \| g) \mathrm{d}x + \left(\int \alpha(x) \mathrm{d}x - 1\right)^2,$$

where the first equality holds by noting that $\int \widetilde{f}(x, \cdot) \mathrm{d}g = 0$ and the inequality follows from the Cauchy-Schwarz inequality. Thus, we complete the proof of Property (iv) of Lemma D.3. $\square$

**Lemma D.4.** For weighted homogeneous Sobolev norm defined by

$$\|\nu_1 - \nu_2\|_{\dot{H}^{-1}(\mu)} = \sup \left\{|\langle f, \nu_1 - \nu_2\rangle| \big| \|f\|_{\dot{H}^1(\mu)} \le 1\right\}.$$

we have the following properties.

(i) For a group of probability measures $\mu_x : \mathcal{X} \to \mathscr{P}_2(\Theta)$ and $\nu_1, \nu_2 \in \mathscr{P}_2(\Theta)$, if $\mu$ is in the convex hull of $\mu_x$, i.e., there exists $\alpha_x \ge 0$ such that both $\int \alpha_x \mathrm{d}x = 1$ and $\mu = \int \alpha_x \mu_x \mathrm{d}x$ hold, it then holds that

$$\|\nu_1 - \nu_2\|_{\dot{H}^{-1}(\mu)} \le \sup_{x \in \mathcal{X}} \|\nu_1 - \nu_2\|_{\dot{H}^{-1}(\mu_x)}.$$

(ii) Assume that we have measures $\mu \in \mathscr{P}_2(\Theta)$ and $\nu_x : \mathcal{X} \to \mathscr{P}_2(\Theta)$. Let $\beta_1, \beta_2 : \mathcal{X} \to \mathbb{R}$ be two functions on $\mathcal{X}$ such that $\int \beta_1(x)\mathrm{d}x = \int \beta_2(x)\mathrm{d}x$. Then, by letting $\nu_1 = \int \nu_x \beta_1(x)\mathrm{d}x$ and $\nu_2 = \int \nu_x \beta_2(x)\mathrm{d}x$, we have

$$\|\nu_1 - \nu_2\|_{\dot{H}^{-1}(\mu)} \le \frac{1}{2} \sup_{(x', x'') \in \mathcal{X} \times \mathcal{X}} \|\nu_{x'} - \nu_{x''}\|_{\dot{H}^{-1}(\mu)} \cdot \int |\beta_1(x) - \beta_2(x)|\mathrm{d}x.$$

*Proof.* **Proof of Property (i) of Lemma D.4.** By definition of the weighted homogeneous Sobolev norm, we have

$$\|\nu_1 - \nu_2\|_{\dot{H}^{-1}(\mu)} = \sup_f \left\{ |\langle f, \nu_1 - \nu_2 \rangle| \,\Big|\, \int |\nabla f|^2 \alpha_x \mathrm{d}\mu_x \mathrm{d}x \le 1 \right\}$$

$$= \sup_{f, \lambda_x \ge 0} \left\{ |\langle f, \nu_1 - \nu_2 \rangle| \,\Big|\, \int |\nabla f|^2 \mathrm{d}\mu_x \le \lambda_x, \forall x; \int \alpha_x \lambda_x \mathrm{d}x = 1 \right\} \quad \text{(D.5)}$$

$$\le \sup_{\substack{\int \lambda_x \alpha_x \mathrm{d}x = 1, \\ \lambda_x \ge 0}} \inf_x \sup_{f_x} \left\{ |\langle f_x, \nu_1 - \nu_2 \rangle| \,\Big|\, \int |\nabla f_x|^2 \mathrm{d}\mu_x \le \lambda_x \right\}, \quad \text{(D.6)}$$

where the first equality holds by noting that $\mu = \int \alpha_x \mu_x \mathrm{d}x$. To illustrate the last equality, we denote by $\mathscr{F}$ and $\mathscr{F}_x$ the allowed function class for $f$ in (D.5) and $f_x$ in (D.6) to choose from, respectively. Note that we have $\mathscr{F} \subseteq \mathscr{F}_x$ for any $x$, which is because the constraints for each $f_x$ in (D.6) are relaxation of the constraints for $f$ in (D.5). And so, the supremum taken over $\mathscr{F}$ is no larger than the supremum over $\mathscr{F}_x$ for any $x$. Therefore, the supremum over $\mathscr{F}$ is no larger than the the smallest supremum over $\mathscr{F}_x$. Thus, (D.6) holds. Furthermore, we have

$$\|\nu_1 - \nu_2\|_{\dot{H}^{-1}(\mu)} \le \sup_{\substack{\int \lambda_x \alpha_x \mathrm{d}x = 1, \\ \lambda_x \ge 0}} \inf_x \sup_{f_x} \left\{ |\langle f_x, \nu_1 - \nu_2 \rangle| \,\Big|\, \int |\nabla f_x|^2 \mathrm{d}\mu_x \le \lambda_x \right\} \quad \text{(D.7)}$$

$$= \sup_{\substack{\int \lambda_x \alpha_x \mathrm{d}x = 1, \\ \lambda_x \ge 0}} \inf_x \left\{ \sqrt{\lambda_x} \|\nu_1 - \nu_2\|_{\dot{H}^{-1}(\mu_x)} \right\} \quad \text{(D.8)}$$

$$\le \sup_{\substack{\int \lambda_x \alpha_x \mathrm{d}x = 1, \\ \lambda_x \ge 0}} \inf_x \left\{ \sqrt{\lambda_x} \right\} \cdot \sup_x \|\nu_1 - \nu_2\|_{\dot{H}^{-1}(\mu_x)},$$

where the first equality holds by noting that (D.8) is a rescaling of (D.7) with respect to $\lambda_x$ in the constraints of (D.7). Here, we let

$$y = \sup_{\substack{\int \lambda_x \alpha_x \mathrm{d}x = 1, \\ \lambda_x \ge 0}} \inf_x \left\{ \sqrt{\lambda_x} \right\}.$$

Then, it holds that $y \le 1$. Otherwise, there must exists $\lambda_x$ such that $\int \lambda_x \alpha_x \mathrm{d}x = 1$ and that $\lambda_x > 1$ holds for any $x \in \mathcal{X}$, which contradicts with our conditions that $\int \alpha_x \mathrm{d}x = 1$ and $\alpha_x \ge 0$. Therefore, it further holds that

$$\|\nu_1 - \nu_2\|_{\dot{H}^{-1}(\mu)} \le y \cdot \sup_x \|\nu_1 - \nu_2\|_{\dot{H}^{-1}(\mu_x)} \le \sup_x \|\nu_1 - \nu_2\|_{\dot{H}^{-1}(\mu_x)}.$$

Thus, we complete the proof of Property (i) of Lemma D.4.

**Proof of Property (ii) of Lemma D.4.** Let $\alpha = \beta_1 - \beta_2$. Then, we have $\int \alpha(x)\mathrm{d}x = 0$. Let $\alpha^+ = \max\{0, \alpha\}$ and $\alpha^- = -\min\{0, \alpha\}$. Then, we have $\alpha^+ - \alpha^- = \alpha = \beta_1 - \beta_2$ and that $\alpha^+ + \alpha^- = |\alpha| = |\beta_1 - \beta_2|$. Since $\int \alpha = 0$, it holds that $\int \alpha^+(x)\mathrm{d}x = \int \alpha^-(x)\mathrm{d}x = A$, where $A \ge 0$. We further let $\lambda(x, x') = A^{-1}\alpha^+(x)\alpha^-(x')$. Then, it holds that $\alpha^+(x) = \int \lambda(x, x')\mathrm{d}x'$ and that $\alpha^-(x') = \int \lambda(x, x')\mathrm{d}x$. Therefore, we have

$$\|\nu_1 - \nu_2\|_{\dot{H}^{-1}(\mu)} = \left\| \int_{\mathcal{X}} (\alpha^+ - \alpha^-)\nu_x \mathrm{d}x \right\|_{\dot{H}^{-1}(\mu)} = \left\| \int_{\mathcal{X} \times \mathcal{X}} \lambda(x, x')(\nu_x - \nu_{x'})\mathrm{d}x\mathrm{d}x' \right\|_{\dot{H}^{-1}(\mu)}.$$

By defintion of the weighted homogeneous Sobolev norm, it further holds that

$$\left\| \int_{\mathcal{X} \times \mathcal{X}} \lambda(x, x')(\nu_x - \nu_{x'}) \mathrm{d}x \mathrm{d}x' \right\|_{\dot{H}^{-1}(\mu)}$$

$$= \sup_f \left\{ \left| \left\langle \int \lambda(x, x')(\nu_x - \nu_{x'}) \mathrm{d}x \mathrm{d}x', f \right\rangle \right| \, \Big| \, \int |\nabla f|^2 \le 1 \right\}. \tag{D.9}$$

We assume the supremum in (D.9) is reached at $f^*$. Then, we have $\int |\nabla f^*|^2 \mathrm{d}\mu \le 1$ and that

$$\left\| \int_{\mathcal{X} \times \mathcal{X}} \lambda(x, x')(\nu_x - \nu_{x'}) \mathrm{d}x \mathrm{d}x' \right\|_{\dot{H}^{-1}(\mu)} = \left| \left\langle \int \lambda(x, x')(\nu_x - \nu_{x'}) \mathrm{d}x \mathrm{d}x', f^* \right\rangle \right|$$

$$\le \sup_{(x,x') \in \mathcal{X} \times \mathcal{X}} |\langle \nu_x - \nu_{x'}, f^* \rangle| \cdot \int \lambda(x, x') \mathrm{d}x \mathrm{d}x'$$

$$\le \frac{1}{2} \sup_{(x,x') \in \mathcal{X} \times \mathcal{X}} \|\nu_x - \nu_{x'}\|_{\dot{H}^{-1}(\mu)} \cdot \int |\beta_1 - \beta_2| \mathrm{d}x,$$

where the last inequality holds by noting that $\int \lambda(x, x') \mathrm{d}x \mathrm{d}x' = A = 1/2 \cdot \int |\alpha| \mathrm{d}x = 1/2 \cdot \int |\beta_1 - \beta_2| \mathrm{d}x$. Thus, we complete the proof of Property (ii) of Lemma D.4. $\qquad \square$

**Lemma D.5.** For probability measures $\nu_1, \nu_2 \in \mathscr{P}_2(\Theta_1)$ and $p_1, p_2 \in \mathscr{P}_2(\Theta_2)$, it holds that

$$W_2^2(\nu_1 \times p_1, \nu_2 \times p_2) \le W_2^2(\nu_1, \nu_2) + W_2^2(p_1, p_2).$$

*Proof.* By the property of optimal transport [6], there exists mapping $T_\nu$ and $T_p$ such that

$$W_2^2(\nu_1, \nu_2) = \int \|\theta_1 - T_\nu(\theta_1)\| \nu_1(\mathrm{d}\theta_1),$$

$$W_2^2(p_1, p_2) = \int \|\theta_1 - T_p(\theta_1)\| p_1(\mathrm{d}\theta_1),$$

and that $(T_\nu)_\sharp \nu_1 = \nu_2$ and $(T_p)_\sharp p_1 = p_2$. Note that we have $(T_\nu \times T_p)_\sharp (\nu_1 \times p_1) = \nu_2 \times p_2$. Thus, by definition of $W_2$ distance, it holds that

$$W_2^2(\nu_1 \times p_1, \nu_2 \times p_2) \le \int \left\| (\theta_1, \theta_2) - \big(T_\nu(\theta_1), T_p(\theta_1)\big) \right\|^2 \nu_1(\mathrm{d}\theta_1) p_1(\mathrm{d}\theta_2)$$

$$= \int \left( \left\| \theta_1 - T_\nu(\theta_1) \right\|^2 + \left\| \theta_2 - T_p(\theta_2) \right\|^2 \right) \nu_1(\mathrm{d}\theta_1) p_1(\mathrm{d}\theta_2)$$

$$= W_2^2(\nu_1, \nu_2) + W_2^2(p_1, p_2).$$

Thus, we complete the proof of Lemma D.5. $\qquad \square$

**Lemma D.6.** For probability density function $\rho, \rho_1, \rho_2 \in \mathscr{P}_2(\Theta)$, let $\widetilde{\rho}_1 = \alpha^{-1}\rho_1 + (1 - \alpha^{-1})\rho$ and $\widetilde{\rho}_2 = \alpha^{-1}\rho_2 + (1 - \alpha^{-1})\rho$. Then, We have

$$W_2(\widetilde{\rho}_1, \widetilde{\rho}_2) \le \alpha^{-1/2} W_2(\rho_1, \rho_2).$$

*Proof.* Recall the definition of Wasserstain-2 distance that

$$W_2(\rho_1, \rho_2) = \left( \inf_{\gamma \in \Gamma(\rho_1, \rho_2)} \int \|x - y\|^2 \mathrm{d}\gamma(x, y) \right)^{1/2},$$

where $\Gamma(\rho_1, \rho_2)$ is the set of all couplings of $\rho_1$ and $\rho_2$. We assume that the infimum is reached by $\gamma^*(x, y) \in \Gamma(\rho_1, \rho_2)$, i.e.,

$$W_2(\rho_1, \rho_2) = \left( \int \|x - y\|^2 \mathrm{d}\gamma^*(x, y) \right)^{1/2}.$$

We denote by $\gamma'(x, y)$ the distribution such that

$$\gamma'(x, y) = \alpha^{-1} \gamma^*(x, y) + (1 - \alpha^{-1}) \rho(x) \delta(x - y),$$

where $\delta(x,y)$ is dirac delta function and it holds that $\gamma'(x,y) \in \Gamma(\widetilde{\rho}_1, \widetilde{\rho}_2)$. Hence, it follows that

$$
\begin{aligned}
W_2(\widetilde{\rho}_1, \widetilde{\rho}_2) &\leq \Big( \int \|x-y\|^2 \mathrm{d}\gamma'(x,y) \Big)^{1/2} \\
&= \Big( \int \|x-y\|^2 \big( \alpha^{-1}\gamma^*(x,y) + (1-\alpha^{-1})\rho(x)\delta(x-y) \big) \mathrm{d}(x,y) \Big)^{1/2} \\
&= \alpha^{-1/2} W_2(\rho_1, \rho_2).
\end{aligned}
$$

Thus, we complete the proof of Lemma D.6. $\qquad\qquad\square$

## E    Auxiliary Lemmas

We use the definition of absolutely continuous curves in $\mathscr{P}_2(\mathbb{R}^D)$ in [6].

**Definition E.1** (Absolutely Continuous Curve). Let $\beta : [a,b] \to \mathscr{P}_2(\mathbb{R}^D)$ be a curve. Then, we say $\beta$ is an absolutely continuous curve if there exists a square-integrable function $f : [a,b] \to \mathbb{R}$ such that

$$
W_2(\beta_s, \beta_t) \leq \int_s^t f(\tau)\,\mathrm{d}\tau
$$

for any $a \leq s < t \leq b$.

Then, we have the following first variation formula.

**Lemma E.2** (First Variation Formula, Theorem 8.4.7 in [6]). Given $\nu \in \mathscr{P}_2(\mathbb{R}^D)$ and an absolutely continuous curve $\mu : [0,T] \to \mathscr{P}_2(\mathbb{R}^D)$, let $\beta : [0,1] \to \mathscr{P}_2(\mathbb{R}^D)$ be the geodesic connecting $\mu_t$ and $\nu$. It holds that

$$
\frac{\mathrm{d}}{\mathrm{d}t} \frac{W_2(\mu_t, \nu)^2}{2} = -\langle \dot{\mu}_t, \dot{\beta}_0 \rangle_{\mu_t, W_2},
$$

where $\dot{\mu}_t = \partial_t \mu_t$, $\dot{\beta}_0 = \partial_s \beta_s \,|_{s=0}$, and the inner product is defined in (A.2).

**Lemma E.3** (Eulerian Representation of Geodesics, Proposition 5.38 in [58]). Let $\beta : [0,1] \to \mathscr{P}_2(\mathbb{R}^D)$ be a geodesic and $v$ be the corresponding vector field such that $\partial_t \beta_t = -\operatorname{div}(\beta_t \cdot v_t)$. It holds that

$$
\frac{\mathrm{d}}{\mathrm{d}t}(\beta_t \cdot v_t) = -\operatorname{div}(\beta_t \cdot v_t \otimes v_t),
$$

where $\otimes$ is the outer product between two vectors.

**Lemma E.4** (Talagrand's Inequality, Corollary 2.1 in [44]). Let $\nu$ be $N(0, \kappa \cdot I_D)$. It holds for any $\mu \in \mathscr{P}_2(\mathbb{R}^D)$ that

$$
W_2(\mu, \nu)^2 \leq 2D_{\mathrm{KL}}(\mu \,\|\, \nu)/\kappa.
$$

**Lemma E.5** (Theorem 1 in [46]). Let $\mu, \nu$ be two probability measures in $\mathscr{P}_2(\theta)$. Then, it holds that

$$
W_2(\mu, \nu) \leq 2\|\mu - \nu\|_{\dot{H}^{-1}(\nu)}.
$$

## F    Conclusions and Limitations

In this work, we study the time envolution of a two-timescale AC represented by a two-layer neural network in the mean-field limit. Specifically, the actor updates its policy via proximal policy optimization, which is closely related to the replicator dynamics, while the critic updates by temporal-difference learning, which is captured by a semigradient flow in the Wasserstein space. By introducing a restarting mechanism, we establish the convergence and optimality of AC with two-layer overparameterized neural network. However, the study has potential limitations. In this work we only study the continuous-time limiting regime, which is an ideal setting with infinitesimal learning rates, and establish finite-time convergence and optimality guarantees. Finite-time results for the more realistic discrete-time setting is left for future research.