# OpenReview forum: "Wasserstein Flow Meets Replicator Dynamics: A Mean-Field Analysis of Representation Learning in Actor-Critic"
_NeurIPS.cc/2021/Conference — NeurIPS 2021 Poster_

### Official Review · Reviewer_ozyQ · 2021-07-14

**Rating:** 6
**Confidence:** 5

**Summary:**

The paper studies convergence of an Actor-Critic algorithm for an MDP problem with infinite one-hidden layer neural network approximation of Q function in the regime mean-field  (Theorem 4.5) and lazy type regime (Theorem 4.6).    By bringing (heavy) machinery from gradient flows literature, the authors perform a similar analysis to the one performed by a group of A. Montanari or Chizat/Bach in the context of supervised learning problem with one-hidden layer approximation which more recently has been adapted by Agazzi an Lu to study popular Policy Gradient algorithm.


**Main Review:**

Originality:  The results seem to be new, and the main contribution is the study of actor-critic algorithm represented by gradient flow equations (3.5) MF-TD and (3.7) MF-PPO. However, the dynamics of 3.7 is very simple and one can see from the error in Theorem 3.7 that the crux of the matter is the estimate on Q functions (where grad flow  3.7 plays a small role).  At that point, it would be very helpful if authors presented a thorough comparison to the work [69] of Zhang et al. and possibly [1] Agazzi et al. Especially result from [69] seems very relevant (see their Section 4), though I only had a chance to skim through that other paper.

Quality: This is theoretical work and as already pointed the results stated in section 4 follow from a heavy analysis of gradient flow. In my opinion, authors did not make sufficient effort to help readers follow the proofs. Even though I'm rather well familiar with the analysis of this type I found reading the paper difficult.

There are a number of things that would require further comments. Let me give a few examples (impossible to list them all here):
- Heuristics in section 3 are helpful, but there isn't any comments on which of these approximation errors would be easy to quantify and which not.
- in 3.3, entropy regularisation appears without any comments.  This doesn't correspond to classical an actor-critic algorithm. Is this needed? How does this help?
-  [top page 7] when giving a reference to technical results, please specify chapter, section, concrete results. Giving a reference to a whole book is useless.
- I have some issues with assumptions 4.2: why the activation function need to be odd? And Lipschtnes in [-2/3,2/3] seems too prescriptive.
- In 4.3: why it is sensible to assume that both reward and transition functions need to admit one-hidden layer representation and with the same activation function? This seems cooked up and little effort has been made to explain why these are sensible assumptions.
- Theorem 4.5 and a discussion that follows: One would expect the bound to decay with t, but instead authors stated results that says that one need to send \eta and \alpha  to \infty. Now \eta in 3.5 corresponds to the time change and \alpha dictates scaling of the neural network approximation. Hence, if I'm reading this correctly, this means that in effect, authors study lazy regime in the end... This is somehow also implied by 'restoring mechanism.'
- Authors mention replicator dynamics and even put this in the title, but there isn't anything in the paper saying why this is relevant.

Clarity: As already stated, I would appreciate more discussion around assumptions and main results and how they relate to what's in the literature. The analysis in the appendix is hard to read, and significantly more effort is needed to make it accessible to NIPS community.

Significance: While I appreciate the importance of this type of work to shed more light on why neural networks based algorithms work, I don't feel this work adds significantly in comparison to what already has been done in the study of on-hidden layer neural networks in mean-field or lazy regimes.

**Time Spent Reviewing:**

3h

---

> ### Author Response · Authors · 2021-08-10
> **Response to Reviewer ozyQ**
>
> We appreciate the valuable reviews and suggestions. We will revise our work accordingly.
>
> - **Significance** Our work is very different than the seminal study of Montanari, Chizat/Bach, and Agazzi and Lu, which focus on the analysis of a single dynamics. In particular, [1, 19, 41, 42] only considers the global optimality of the gradient flow, and [18] considers the non-asymptotic convergence of the gradient flow in the lazy regime. In stark contrast, our paper considers the coupling of two dynamics: MF-TD (Wasserstein _semi-gradient flow_) and MF-PPO (replicator dynamics). The coupling of the two dynamics makes the problem significantly more challenging. To name a few, one significant challenge comes from the proof of Lemma B.1. The proof utilizes the techniques of mollifiers and the equivalence of the Wasserstein distance to negative order Sobolev norm, which are novel applications of such mathematical techniques to the study of RL with two-layer neural networks. Moreover, we provide a detailed comparison of our work to [1, 69] in the following discussion.
>
> - **Dynamics (3.7)** The key challenge of our proof comes from the coupling of the TD and PPO dynamics. When we have access to the true action-value function, the proof will be much easier. However, as TD and PPO are coupled, analyzing the policy evaluation error becomes significantly much more challenging than a vanilla policy evaluation problem.
>
> - **Comparison with [1] Agazzi et al.** [1] mainly focuses on the global optimality of softmax policy gradient with neural networks. In particular, their study differs from ours in the following perspectives.
>
>   - [1] considers the softmax policies and entropy-regularized reward. In contrast, we do not specify the parameterization of the policy (by updating the policy by the closed-form of PPO) and do not require entropy regularization.
>
>   - [1] focuses on the global optimality softmax policy gradient methods. They show that _once the policy gradient flow converges_, it must converge to the global optimal solution. In contrast, we show a non-asymptotic convergence of two-timescale AC dynamics to the global optimal solution.
>
>   - [1] assumes access to the true action-value function $Q^\pi$ for each policy. In contrast, we consider an actor-critic algorithm where the action-value function is learned (and is not learned exactly or with high precision). As we stated, the coupling of TD and PPO makes our study much more challenging.
>
> - **Comparison with [69] Zhang et al.** [69] mainly focuses on the policy evaluation problem, while our paper focuses on the actor-critic algorithm, which considers a policy improvement problem with policy evaluation as a subproblem.
>
> - **Approximation error** We remark that taking the continuous-time limit is an important technical tool for the analysis of two-time scale algorithms ([13] Borkar, 2009). Also, the error of taking infinite-width and continuous limit has been characterized when considering TD and PPO as two decoupled dynamics. Specifically, the approximation error of temporal difference learning (or SGD) to its infinite-width, continuous-time limit has been characterized by [41-42] Mei et al. (2018, 2019) and [69] Zhang et al. (2020). The approximation of PPO to its continuous-time limit is characterized in L.201-L.204. However, as stated in the limitations (Appendix F), when the updates of TD and PPO are coupled with each other, analyzing the approximation error is much more challenging but is possible by existing techniques. On the other hand, in the empirical study of actor-critic, the step-size is always set to be small (e.g. learning_rate = 0.0007 in [A2C of Stable Baselines 3](https://stable-baselines3.readthedocs.io/en/master/modules/a2c.html)) and thus, approximates the continuous-time limit. The scale of the neural network is always very large and approximates the infinite-width limit (generally empirical studies consider deeper neural networks but here we only consider two-layer neural networks).
>
> - **Entropy regularization in (3.3)** The entropy regularization term in (3.3) appears in the standard PPO update ([49] Schulman et al.), where the intuition comes from the mirror descent. In particular, the update in (3.3) is equivalent to $$\pi_{k+1} = \arg\max_\pi \Bigl\{ \underbrace{\langle Q_k, \pi - \pi_k\rangle}_{\mathrm{(i)}} - \underbrace{\epsilon^{-1} \mathrm{KL}(\pi \,\|\, \pi_k) \Bigr\}}_{\mathrm{(ii)}}.$$​ As the performance difference lemma (Theorem 6.2 in [33]) suggests that term (i) is a local estimation of $V^{\pi}$​ and term (ii) regularizes the policy update so that the local estimation is of high precision. We will add a detailed discussion of the KL regularization in our revision.
>
> - **Specify the reference** We will add the detailed reference for those results. Specifically, the uniform function approximation theorem can be found in Theorem 3.1 of [47]. The regularity issue for Wasserstein space over manifolds with boundary can be found in e.g. Example 14.10 and Remark 17.32 of [59]. In general, the study of Wasserstein space over manifolds with boundary is limited and thus, we use the $\beta$ function to avoid such cases.
>
> - **Assumption 4.2** As stated in L.261-L.262, such assumption is satisfied by many common activation functions, for example, $\beta = B \cdot \tanh$ and $\tilde \sigma = \tanh$. This assumption of the oddness of $\tilde \sigma$ is required for the generality of the transition kernel structure in (4.6) and is used for proving the equivalence of (4.6) to the class defined between L.280 and L.281. In particular, the condition that $\varphi$ should be positive makes the structure more restricted. Thus, by assuming the oddness of $\tilde \sigma$, the class of (4.6) is equivalent to the class where $\varphi$ need not be nonnegative (as in the class defined between L.280 and L.281). In addition, the oddness of $\beta$ does not lose any generality since the corresponding function class is equivalent to (4.3). On the other hand, the Lipschtnes-continuity of $\beta^{-1}$ in $[-2/3,2/3]$ can be generalized to the condition that $\beta^{-1}$ is Lipschitz continuous on a compact sub-interval of $(-1, 1)$ around $0$. Taking $\tanh$ as an example, we see that $\tanh^{-1}$ is not Lipschitz continuous in $(-1, 1)$ but is Lipschitz continuous in the compact sub-intervals. We describe the condition in such a way so as to simplify the presentation of the proof and theorem (since we need to characterize the dependency on $\ell_\beta$).
>
> - **Transition and reward admit the same activation** Following the universal function approximation theorem for neural networks, the function class captured by neural networks is rich enough to capture a large class of practical MDPs. Also, see L.281-L.283. In our paper, we assume that there exists an absolute constant bound of the output weights to capture the MDP in our paper. Thus, the structure assumption is reasonable as most MDPs can be approximated by such a structure with sufficiently large output weights.
>
> - **Relation to the Lazy regime** In the common NTK regime, $\alpha$ is set to be $\sqrt{m}$ where $m$ goes to infinity. We allow $\alpha$ to be a large constant and tunable, which thus lies in between the NTK regime and the mean-field regime. As stated in L.183-184, our parameterization of the critic is similar to the lazy regime. However, our algorithm consists of two parts, PPO for the actor and TD for the critic, and the coupling makes our algorithm very different than the lazy regime. As we stated above, the coupling of Wasserstein semi-gradient flow (MF-TD) and replicator dynamics makes the analysis significantly more challenging.
>
> - **Replicator dynamics** The replicator dynamics originally arises in the study of evolutionary game theory ([^1] Schuster et al. 1983). In brief, for a given function $f$, the replicator dynamics is given by the differential equation $$\frac{d}{dt}x_t(a) = x_t(a) [f(a, x_t) - \phi(x)],$$ where $\phi(x) = \int x(a)f(a, x)$. Then, for a fixed $s$, let $x(a) = \pi(a \,|\, s)$ and $f(a, x) = Q^\pi(s, a)$, we see that (3.7) corresponds to a replicator dynamics if $Q_t = Q^{\pi_t}$. Note that in our paper, we do not have access to the true action-value function $Q^\pi$. Thus, we use the estimator $Q_t$ calculated by the critic step. We will add a detailed introduction to replicator dynamics in our revision.
>
> - **Organizing the proof** We will add a proof sketch of the intuition behind our proof in the revision. Moreover, besides the mean-field-based analysis, our proof also utilizes the techniques in the analysis of PPO and the two-timescale dynamics in bilevel optimization. We will add an introduction to PPO, actor-critic, and bilevel optimization to make the proof easier to understand in our revision.
>
> [^1]: Schuster, Peter, and Karl Sigmund. "Replicator dynamics." Journal of theoretical biology 100.3 (1983): 533-538.

---

> > ### Comment · Reviewer_ozyQ · 2021-08-24
> > **Discussion.**
> >
> > Thank you for carefully addressing my concerns. My overall assessment of the paper remains unchanged.

---

### Official Review · Reviewer_yWAT · 2021-07-15

**Rating:** 6
**Confidence:** 3

**Summary:**

This paper analyzes the optimality and convergence of the two-scale update for an actor-critic algorithm where the actor and critic are parameterized by two layer NNs (which is a generalization of traditional linear function approximation guarantees in RL). The authors analyze for the PPO as actor and minimizer of the the MSBE as the critic. Considering the actor-critic updates as gradient flow differential equations in the L_2 Wasserstein space, in following steps, the authors show: (1) optimality and convergence rate of policy error proportional to inverse of time and the critic error (2) bounding the critic error based on the Wasserstein error of the distribution of NN weights (3) applying (1) and (2) with Restarting mechanism (resampling the critic weights from an initial distribution when they are sufficiently close to the initial distribution), they prove the global optimality and convergence rate of AC method .

**Limitations And Societal Impact:**

They discussed potential analytical limitations in the conclusion in the appendix

**Main Review:**

The paper is clear and enjoyable to read. The idea of the paper is interesting and the convergence analysis of an actor critic algorithm with two layer neural networks potentially helps strengthening RL theoretical foundation.

My main concern about this paper is the update 3.2 that is a based on minimizing MSBE which suffers from double sampling and I couldn’t find it being incorporated/or discussed in the analysis of the paper. I am interested in knowing if and how much this affects the convergence analysis in this paper?

Line 276: does not lose…

**Time Spent Reviewing:**

4

---

> ### Author Response · Authors · 2021-08-10
> **Response to Reviewer yWAT**
>
> We appreciate the valuable reviews and suggestions. We will revise our work accordingly.
>
> - **Double sampling** Minimizing MSBE by _vanilla_ stochastic gradient descent suffers from double sampling. That is why we minimize MSBE via temporal difference learning, which updates the parameter along the direction of stochastic semi-gradient. Such an algorithm is known to tackle the double sampling issue in minimizing MSBE. Specifically, temporal difference learning updates the parameter via $$\theta'=\theta - \epsilon \cdot (Q_\theta(s, a) - r(s, a) - \gamma Q_\theta(s', a')) \cdot \nabla_\theta Q_\theta(s, a),$$ which does not require the independent samples of $(s', a')$ and thus, avoid the double sampling issue.
> See, e.g., Section 2.2.1 in Dann et al., 2014[^1] for a detailed discussion.
>
> [^1]: Christoph Dann, Gerhard Neumann, Jan Peters, 2014, Policy Evaluation with Temporal Differences:
> A Survey and Comparison, JMLR

---

### Official Review · Reviewer_R6mo · 2021-07-15

**Rating:** 7
**Confidence:** 2

**Summary:**

This paper studies actor-critic (AC) algorithms with neural networks from a theoretical perspective.

The authors consider an AC algorithm variant that updates its actor with PPO with a small stepsize, and its critic updated via TD and a larger stepsize. Under certain conditions, the authors prove that this AC reaches the optimal policy with sublinear rate. The authors additionally show that the feature representations learned by the critic can move further away from the initialization.

**Limitations And Societal Impact:**

Since the authors are analyzing neural AC algorithms, it might be really interesting to have some numerical experiments, especially on the relative learning rate of actor and critic. And it would be interesting to have analysis on how this relates to other problems in RL such as bias accumulation. However, the theoretical results in this paper seem to be already quite important.

**Main Review:**

- Originality: authors approached the neural AC problem from very interesting angles and the approach as well as the results seem to be novel
- Quality: The theoretical results in the paper seem interesting and important
- Clarity: the paper is clearly written
- Significance: significant results that improves understanding of how actor-critic methods learn with neural networks

Questions/comments for the authors:
- Why would we want to update the critic much faster than the actor? In many recent paper on deep reinforcement learning (for example, TD3, SAC), the actor and critic have similar or same learning rates. And if the two are updated at the same rate, how would that affect the results proven in the paper?
- The results on representation can evolve into a larger neighborhood seems to be very interesting, but why is this significant? What does it tell us about how actual neural networks should be treated differently from NTK? In my opinion it would be great if the authors can provide some more discussion here.
- The paper mentions many times the close relationship to replicator dynamics, but it seems to me the paper does not have a good background introduction to exactly what this relationship is, and this might be a bit confusing for readers who are not that familiar with replicator dynamics.

Minor comments:
- there is a typo at line 178 "the the critic"

==================

post rebuttal: I have read the responses from the authors, I feel my questions are addressed and my evaluation remains the same

**Time Spent Reviewing:**

3

---

> ### Author Response · Authors · 2021-08-10
> **Response to Reviewer R6mo**
>
> We appreciate the valuable reviews and suggestions. We will revise our work accordingly.
>
> - **Two timescale update** Updating the _parameter_ of the actor and the critic at the same learning rate may not mean updating the _function_ of the actor and the critic at the same rate. That is because that the scale of the actor and the critic may not be the same. For example, let the actor to be $\pi = \phi_\theta$ and the critic to be $Q = \alpha \phi_w$ for a large constant $\alpha$. We consider the mean-squared loss for $Q$. When both $\theta$ and $w$ moves one step with rate $\epsilon$, we see that $\pi$ moves $\epsilon$ while $Q$ moves $\alpha^2 \epsilon$ (due to $\nabla (Q - y)^2 = (Q - y) \cdot\nabla Q = O(\alpha^2)$). Despite the simplicity of the example, we see that the critic is updated much faster than the actor although their parameters are updated at the same rate. We remark that the TD update of in our paper is normalized to cancel the scaling effect of $\alpha$, that is, the update in (3.2) is derived from $\theta' = \theta - \eta\epsilon /\alpha^2 \cdot \text{semi-grad}(\text{MSBE}(Q_\theta))$. Thus, the relative timescale $\eta$ is the learning rate of the function of the critic with respect to the actor. In fact, our paper allows that the parameters of the actor and the critic are updated at the same rate.
>
> - **Significance of the evolution of the representation** In the common NTK regime, $\alpha$ is set to be $\sqrt{m}$ while $m$ goes to infinity. Thus, the NTK-based analysis only allows an infinitesimal deviation of the feature representation with respect to the initial one (also see L.327-L.333). In contrast, our study allows an $O(\alpha^{-1})$ deviation of the feature representation, where $\alpha$ is tunable. The significance of the evolution is also introduced in L.18-L.39. In brief, the empirical success of DRL is significantly enhanced by the power of neural networks to learn data-dependent feature representation. However, common NTK-based analysis essentially considers data-independent feature representation since the feature representation is in an infinitesimal neighborhood of the initialization.
>
> - **Replicator dynamics** The replicator dynamics originally arises in the study of evolutionary game theory ([^1] Schuster et al. 1983). In brief, for a function $f$, the replicator dynamics is given by the differential equation $$\frac{d}{dt}x_t(a) = x_t(a) [f(a, x_t) - \phi(x)],$$ where $\phi(x) = \int x(a)f(a, x)$. Then, for a fixed $s$, let $x(a) = \pi(a \,|\, s)$ and $f(a, x) = Q^\pi(s, a)$, we see that (3.7) corresponds to a replicator dynamics if $Q_t = Q^{\pi_t}$. Note that in our paper, we do not have access to the true action-value function $Q^\pi$. Thus, we use the estimator $Q_t$ calculated by the critic step. We will add a detailed introduction to replicator dynamics in our revision.
>
> - **Numerical Experiments** We remark that our paper is focused on the analysis of the limiting case of an existing algorithm (actor-critic) via a new technical tool. We will rerun some standard baselines (e.g. A2C,  PPO) to validate our theory.
>
> **Additional Reference**
> [^1]: Schuster, Peter, and Karl Sigmund. "Replicator dynamics." Journal of theoretical biology 100.3 (1983): 533-538.

---

> > ### Comment · Reviewer_R6mo · 2021-08-31
> > **Thank you for the response**
> >
> > Thank you for the detailed response! My evaluation remains the same.

---

### Official Review · Reviewer_5c4N · 2021-07-16

**Rating:** 6
**Confidence:** 3

**Summary:**

The authors analyse a mean-field limit of an actor-critic algorithm for which the critic is parametrized with a single-hidden-layer neural network, and the policy is tabular. The analysis is carried out in continuous time, and in the limit of infinite network width.

**Limitations And Societal Impact:**

As mentioned above, more discussions of the assumptions made for the main theoretical results (including their limitations) would be useful.

**Main Review:**

The analysis of reinforcement learning algorithms under non-linear function approximation is an important direction for better understanding the behavior of deep reinforcement learning agents. As such, the topic the authors study is likely to be of interest to the (theoretical side of) the NeurIPS RL community.

The mean-field limit considered here is interesting and elegant, and broadly the mathematical presentation is precise. A wide variety of related work is presented. However, at the moment I think the paper is currently lacking a broader discussion of the context of these results. This is the primary reason for the current rating for the paper, and I would be open to revising the authors if the authors can address these concerns.

At present, a lot of space is given to detailing aspects of the convergence proof, at the expense of e.g. a conclusion at the end of the main paper, and more discussion/motivation of the various limits described in Section 3, and what these results have to say about Q-learning with neural networks. In particular, I was left with several questions about the motivation and utility of the main results:
 * To obtain the mean-field limit Eqns (3.5) and (3.7), several limits are taken (infinite width, and stepsizes/continuous-time updates). I was unsure whether the order in which these limits are taken affects the analysis, and how closely we should expect the analysis of these systems to relate to finite-width networks updated in discrete-time, and what the analysis can tell us about these systems. Some additional discussion of these connections would be useful, and small-scale experiments to understand when the limit is likely to "kick in" may also be useful.
 * There are some seemingly strong assumptions made in the course of the analysis, such as a tabular policy, bounded output weights for the neural network in Eqn (4.2), structure assumptions on the MDP (Assumption 4.3), and an algorithmic restarting mechanism, presumably to avoid divergence. I would expect these to limit the applicability of the main theoretical result, but there is not much discussion of the practical implication of these assumptions in the paper.

Minor technical comments

Why can we assume (s, a) has uniformly bounded norm without loss of generality? It seems this induces some topological assumptions on S x A (precompactness, for example) - can the authors make clear why their arguments would still apply to state-action spaces which aren't precompact? Is the argument at Equation (4.4) affected by this?

Under the general assumptions on the structure of state-action space, I don't think it holds that Q^pi is the unique minimizer of MSBE - some kind of qualification allowing  \tilde{\Phi}-almost sure equality is required (i.e. consider modifying Q^pi at a single state-action pair that has zero probability under \tilde\{Phi}^pi).

Line 245: Is there a typo here? Should (-B_beta, B_beta) be [-B_beta, B_beta]?

Line 248: "Such over-representation appears to be essential for our analysis". Can more be said about this?

---

AFTER REBUTTAL:

I thank the authors for their clarifications on many of the questions I mentioned in the original review, and have increased my rating for the paper. There are several points of discussion (for example, the boundedness of state-action features) here that would benefit the paper to be included in later versions.

**Time Spent Reviewing:**

5

---

> ### Author Response · Authors · 2021-08-10
> **Response to Reviewer 5c4N**
>
> We appreciate the valuable reviews and suggestions. We will revise our work accordingly.
>
> - **Broader discussion** As mentioned in the introduction, our paper is motivated by the gap between the empirical success of actor-critic type algorithms with neural networks and the limited theoretical understanding of such algorithms. Existing NTK-based methods do not allow the feature representation to go beyond the initial point, which makes these methods unable to explain the success of feature learning in deep reinforcement learning. Thus, we employ the novel mean-field analysis to provide a theoretical understanding of actor-critic with neural networks.
>
> - **Continuous-time, infinite-width limit** We remark that taking the continuous-time limit is an important technical tool for the analysis of two-time scale algorithms ([13] Borkar, 2009). Also, the error of taking infinite-width and continuous limit has been characterized when considering TD and PPO as two decoupled dynamics. Specifically, the approximation error of temporal difference learning (or SGD) to its infinite-width, continuous-time limit has been characterized by [41-42] Mei et al. (2018, 2019) and [69] Zhang et al. (2020). The approximation of PPO to its continuous-time limit is characterized in L.201-L.204. However, as stated in the limitations (Appendix F), when the updates of TD and PPO are coupled with each other, analyzing the approximation error is much more challenging but is possible by existing techniques. On the other hand, in the empirical study of actor-critic, the step-size is always set to be small (e.g. learning_rate = 0.0007 in [A2C of Stable Baselines 3](https://stable-baselines3.readthedocs.io/en/master/modules/a2c.html)) and thus, approximates the continuous-time limit. The scale of the neural network is always very large and approximates the infinite-width limit (generally empirical studies consider deeper neural networks but here we only consider two-layer neural networks). Besides, the focus of our paper is providing theoretical analysis for existing algorithms by considering the limiting case. And we plan to rerun some baselines (e.g., those in [^1] Bjorck et al. 2021) to validate our theory and include the empirical results in our full version.
>
> - **Tabular policy** We remark that we are not using the tabular policy. Instead, our study allows for continuous state-action space. As explained in L.201-L.206, the dynamics of MF-PPO $\frac{d}{dt}\pi_t = \pi_t A_t$ is the continuous-time limit of the PPO dynamics in (3.3) when we have access to the exact solution to the maximization subproblem in PPO (Eqn. 3.3). In practice, PPO solves the maximization subproblem to high precision ([49] Schulman et al., 2017). Thus, we take MF-PPO as a lens for the analysis.
>
> - **Bounded output weights and structural assumption** Following from the universal function approximation theorem for neural networks, when the upper bound of the output weights is sufficiently large, the function class captured by neural networks is rich enough to capture a large class of practical MDPs. Also, see L.281-L.283. Thus, the structure assumption is reasonable as most MDPs can be approximated by such a structure with sufficiently large output weights.
>
> - **Restarting mechanism** As stated in Theorem 4.6, such restarting mechanism helps the convergence of the two-timescale AC by stabilizing the dynamics. In practice, restarting should be imposed when the distance between the iterates and the initialization passes some threshold. Besides, as stated by the equation between L.344 and L.345, for any given time-period $T$, the total restarting number $N$ is finite. This shows that it is easy to implement the restarting mechanism in practice.
>
> - **Uniformly bounded norm** We agree that under the assumption of the uniformly upper-bounded norm, the state-action space is precompact since it is a bounded subset of the Euclidean space. However, such an assumption is common in RL (e.g., Atari and Go). As long as the state-action space is bounded, we can normalize the space to satisfy $\| (s, a)\| \le 1$. There exist scenarios where the state-action space is not bounded. However, the theoretical analysis for such scenarios relies on properties such as the state-action space is upper-bounded with high probability (e.g., LQR with Gaussian noise) and the existence of an upper-bounded feature mapping. Thus, such settings can be converted to the setting of the uniformly upper-bounded state-action space. Furthermore, the uniformly upper-bounded norm of the state-action space is required by (4.4), since $\|\nabla_w \sigma(w^T x)\| = \|\sigma'(w^Tx)\| \cdot \|x\|$. If we assume that (4.4) holds, we can drop the assumption on the uniformly upper-bounded state-action space. In general, when $S\times A$ is unbounded, the learning problem can be information-theoretically infeasible to solve.
>
> - **Unique minimizer of MSBE** As stated in L.156, the weighting distribution $\tilde \Phi^\pi$ has full support, which ensures that the global minimizer is unique in the $L_2$ space over $S\times A$ with the Lebesgue measure. (Note that by definition the $L2$ space modules the equivalence relation of almost everywhere equality.) Besides, under structural assumption and the assumption on the activation function, the function $Q$ and $r$ are both continuous, which ensures that the solution is unique and that modifying a single state-action value is not possible. We will add a detailed description in our revision.
>
> - **Typos** We remark that $(-B_\beta, B_\beta)$ is not a typo. As stated in L.242, the function $\beta$ takes value in $(-1, 1)$, which makes the $\beta'$ takes value in $(-1, 1)$ rather than $[-1, 1]$.
>
> - **Over-representation** The rescaling of $\alpha$ helps us to control the distance between $\rho_0$ and $\rho_t$. In detail, Lemma B.1 (iii) shows that the Wasserstein distance between $\rho_0$ and $\rho_{\pi_t}$ is upper bounded by $O(1/\alpha)$. We will add the discussion on the intuition behind the proof.
>
> [^1]: Bjorck, Johan, Carla P. Gomes, and Kilian Q. Weinberger. "Towards Deeper Deep Reinforcement Learning." arXiv preprint arXiv:2106.01151 (2021).

---

### Decision · Program_Chairs · 2021-09-27

**Decision:**

Accept (Poster)

**Comment:**

All reviewers thought this paper made useful contributions around strengthening the theoretical foundations of RL through an analysis of an AC algorithm with neural networks. AC is a very popular approach in RL and this will likely be of interest to the large body of researchers interested in RL. The authors are encouraged to address the reviewers' feedback in their camera ready paper.